# Graph Neural Network Is A Mean Field Game

## Abstract

In current graph neural networks (GNNs), it is a common practice to apply a pre-defined message passing heuristics to all graph data, even though the stereotypical relational inductive bias (e.g., graph heat diffusion) might not fit the unseen graph topology. Such gross simplification might be responsible for the lack of an in-depth understanding of graph learning principles, which challenges us to push the boundary from crafting application-specific GNNs to embracing a "meta-learning" paradigm. In this work, we ratchet the gear of GNN another notch forward by formulating GNN as a *mean field game*, that is, the best learning outcome occurs at the *Nash*-equilibrium when the learned graph inference rationale allows each graph node to find what the best feature representations is for not only the individual node but also the entire graph.

We formulate the search for novel GNN mechanism into a variational framework of *mean-field control* (MFC) problem, where the optimal relational inductive bias is essentially a graph mapping of *control patterns* associated with mean-field information dynamics. Specifically, we seek for the topology-adaptive control function of transportation mobility (controlling information exchange throughout the graph) and reaction mobility (controlling feature representation learning on each node), on the fly, which allows us to uncover the most suitable learning mechanism for a GNN instance by solving an MFC variational problem through the lens of *Hamiltonian flows*. In this context, our variational framework brings together existing GNN models into various mean-field games with distinct equilibrium states, each characterized by the learned control patterns. Furthermore, we present an agnostic end-to-end deep model, coined *Nash-GNN* (in honor of Nobel laureate Dr. John Nash), to jointly identify the inductive bias and fine-tune the GNN hyperparameters on top of the elucidated learning mechanism. *Nash-GNN* has achieved SOTA performance on diverse graph data including popular benchmark datasets and human connectomes. More importantly, the mathematical insight of MFC framework provides a new window to understand the foundational principles of graph learning as an interactive dynamical system, which allows us to reshape the idea of designing next-generation GNN models.

## 1    Introduction

We live in a world of complex systems where individual objects are intricately connected (Zeng et al., 2017). Since graph is a powerful object to model the object-to-object relationship in the complex system (Smith & Johnson, 2020), graph neural networks (GNNs) that perform machine learning on graph data have been successfully deployed in various fields of science and engineering, including social network analysis (Hamilton et al., 2017a), recommendation systems (Ying et al., 2018), biochemical engineering (Kearnes et al., 2016), knowledge graph (Schlichtkrull et al., 2018), traffic flow prediction (Ma et al., 2020), drug repurposing (Goh et al., 2017), etc.

Despite various GNN instances, their machine learning backbones share many common components. For example, the learning process typically consists of (1) a feature representation module for individual graph nodes and (2) a message passing mechanism to disseminate these feature representations across the graph (Wu et al., 2020). In addition, the inductive biases – *In the realm of GNN, relational inductive bias refers to the inherent assumptions and biases (e.g., permutation invariance and message passing) in the model design that leverages the structure and properties of graphs to improve learning and generalization.* – are critical in all GNNs which are closely associated with

graph topology (Battaglia et al., 2018). That is, the relational inductive biases should preserve explicit relationships between graph nodes (encoded in an adjacency matrix), regardless of using graph convolution network (GCN) (Kipf & Welling, 2017) or Transformer backbone (Ma et al., 2023).

Meanwhile, the complexity of real-world graph data presents new challenges. For instance, multiple lines of evidence show even simple multi-layer perceptrons (MLPs) can outperform GNNs on heterophilic graphs (Zhu et al., 2020; Liu et al., 2021; Mao et al., 2024), where the homophily assumption (i.e., strongly connected nodes are supposed to bear similar feature representations (Kipf & Welling, 2017)) does not hold anymore. Tremendous efforts have been made to address this challenge, where representative approaches include extending single channel message aggregation to adaptive channel mixing (Luan et al., 2022) and directly measuring the degree of graph heterophily (Zhu et al., 2021). Another well-known challenge in GNNs is the over-smoothing issue, due to excessive information aggregation between inter-connected nodes. A set of effective solutions have been proposed in the past several years. For example, graph neural diffusion (GRAND) (Chamberlain et al., 2021) alleviates the over-smoothing issue by replacing the layer-by-layer optimization in the (discrete) GNN model with a continuous diffusion process which is formulated in a partial differential equation (PDE). This approach has inspired the development of several new GNN models (Wang et al., 2022; Rusch et al., 2022; Choi et al., 2023; Thorpe et al., 2022; Brandstetter et al., 2021; Eliasof et al., 2021), which are essentially derived from discretizations of different PDEs. Moreover, adjusting the network architecture provides an additional solution for the over-smoothing issue. For example, ResNet-based machine learning backbones have been integrated into GNNs (Li et al., 2018; 2019), alongside the adoption of Transformer backbone to efficiently capture global feature representations without the restriction of step-by-step traversal on the graph (Kim et al., 2022; Yun et al., 2019).

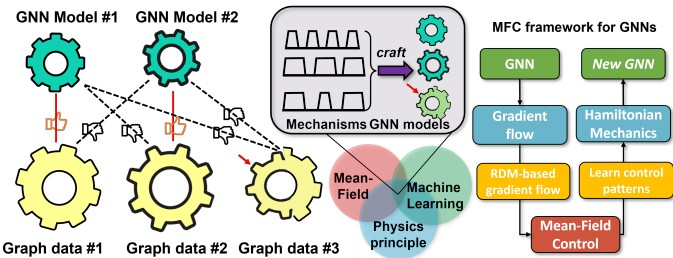

As the mechanic example of gear-to-gear matching shown in Fig. 1, the state-of-the-art behind the design of each GNN model is hand-crafted based on the domain knowledge of machine learning and graph signal processing (Ortega et al., 2018; Hammond et al., 2011). More critically, the design of GNN model is fixed for all graph data, and completely relies on back-propagation to optimize the learning performance by fine-tuning the model hyperparameters. Due to the lack of a good understanding about the learning mechanism inside GNNs at a system level, achieving state-of-the-art performance doesn't necessarily ensure that the inference principle is well-suited for the underlying graph learning problem. A typical example is that a particular GNN model either yields superior performance on homophilic graphs compared to heterophilic ones, or vice versa. In this regard, it is vital to unify the existing work of GNNs into a novel learning paradigm that crafts explainable GNNs with well-defined principles and mathematically guaranteed foundations.

Figure 1: The motivation of *Nash-GNN*. GNN instances and graph data intended for application are visually represented by gears of two different colors (green for models and yellow for data). The tooth pattern of data gears reflects the topological characteristics of graph data, such as whether it exhibits homophilic or heterophilic behavior. *Left*: Current approaches use the same GNN gear to match all data gears, relying on the "black-box" of gradient descent to fine-tune the underlying green gear (GNN model) to fit the yellow gears (data). This often leads to sub-optimal learning outcomes. *Right*: Our MFC framework formulate graph learning as a mean-field game in the variational framework. By formulating the graph feature representation learning as a dynamical process, we capitalize on the equilibrium state (critical point of the mean-field control problem) to form a Hamiltonian flow which allows us to link the "on-the-fly" design of GNN mechanism (shown in the blue box) and the discrete GNN instance for the real-world graph data. The showcase of our *Nash-GNN* approach is the craft of suitable GNN mechanism (indicated by red arrows) for unseen graph data #3.

To establish the physics-informed framework generalizing GNNs instances, we conceptualize the learning process in GNNs as a dynamical system of a large number of particles (i.e., graph nodes), where each particle solves its own feature representation learning problem by taking other particles' learning results into account. This conceptualization is supported by recent works that frame the feature representation learning as a dynamic process of graph heat diffusion (Chamberlain et al., 2021; Dan et al., 2023). By regarding the feature vector on each node as the potential energy, we further hypothesize that the system dynamics follows *the second law of thermodynamics* (Onsager,

1949): energy tends to evolve in the direction where the negative entropy functional dissipates most rapidly. This dissipative nature is given by Onsager's reciprocal relations (Onsager, 1949), simply stating that "Heat does not spontaneously flow from a colder body to a hotter body (Cengel & Boles, 2010)".

Following this notion, we propose to characterize the collective behaviors of simultaneous message passing and feature representation learning within GNNs based on the second law of thermodynamics, where the energy dissipation pattern is determined by a selected (learnable) "free energy functional". Inspired by the *mean-field* theory in physics (Lasry & Lions, 2007), we further conceptualize that the dynamic process of graph feature representation learning forms a gradient flow controlled by the graph-adaptive "free energy functional". In this context, the most promising learning mechanism (aka. relational inductive bias in GNNs) for the specific graph data is characterized by an *optimal transport* from the initial feature representations to an *Nash*-equilibrium state (depending on the down-stream learning tasks), where each graph node finds the best feature presentation for itself and the entire graph.

Taken together, we present a variational framework of *mean-field control* (MFC) problem to achieve the physics-informed learning paradigm of designing novel GNN mechanisms, where the optimal relational inductive bias is essentially the mean-field control pattern associated with underlying graph data. Specifically, the MFC control pattern consists of two functions of kinetic dynamics: (1) a function of transportation mobility for controlling node-to-node information exchange and (2) a function of reaction mobility for learning the feature representation on each node. These two mobility functions determine the characteristic control pattern that yields relation inductive bias (message passing and representation learning) within the specific GNN instance, by shaping the kinetic dynamics of underlying "free energy functional". Since different graph data forms a unique dynamical system with distinct transportation and reaction functions, our proposed physics-informed learning framework for GNN, coined as *Nash-GNN* (name after *Nash* equilibrium), integrates two hierarchical machine learning modules at both the mechanism and model instance levels. At the mechanism level, we seek the most suitable control patterns that shape the gradient flow in the MFC variational problem. Furthermore, we derive *Hamiltonian flows* for the governing equation of the underlying MFC problem. As multiple lines of work have demonstrated that GNN is equivalent to an underlying PDE (Zhao et al., 2024; Dan et al., 2023; Chamberlain et al., 2021; Wang et al., 2022), the Hamiltonian flow becomes a stepping stone, which allows us to link the abstract learning of GNN mechanism in the variational framework and the fine-tuning hyper-parameters at the GNN model instance level. The outcome of our work is an end-to-end deep model that jointly identifies the most suitable relational inductive bias and refines hyper-parameters for the corresponding GNN instance.

The major technical contributions are three-fold. *First*, we present a physics-informed learning framework for GNN that crafts the most suitable GNN model while performing machine learning on graph data. *Second*, we integrate the theory of mean-field control into graph neural networks which not only offers an in-depth understanding of GNNs but also provides a general guideline for developing deep models for unseen graph data. *Third*, we present a practical end-to-end solution, based on Hamiltonian flows, to customize the best GNN model for the underlying graph data. In addition to the comprehensive evaluation on benchmark datasets, we explore the foundational principles of graph learning as an interactive dynamical system, which is valuable for the conceptual framework of developing future GNN models.

## 2 METHODS

### 2.1 BACKGROUND AND MOTIVATION

Suppose we have an undirected, weighted graph $\mathcal{G} = (\mathcal{V}, \mathcal{P})$ with $\mathcal{V} = \{v_i | i = 1, ..., N\}$ is a finite set of $N$ vertices and $\mathcal{P} \subset \mathcal{V} \times \mathcal{V}$ denotes the set of edges. The adjacency matrix is denoted as $A = [a_{ij}]_{i,j=1}^N$, where $[i, j] \in \mathcal{P}$. Suppose $x_t = \{x(v, t) | v \in \mathcal{V}\} \in \mathbb{R}^{N \times d}$ represent the distribution of graph feature embeddings associated at time $t$. The continuity equation describes the evolution of distribution $x_t$ can be formulated as $\frac{\partial}{\partial t} x_t = -div(x_t \gamma_t)$, where $div(\cdot)$ denotes divergence operator and $\gamma_t$ is the latent velocity field. By constraining the evolution of $x_t$ being the gradient flow that minimizes the energy functional $\mathcal{E}(x_t) = \int_{\mathcal{G}} G(x_t) dv$, there exists a unique potential function $\Phi_t$ such that $\gamma_t = \nabla \Phi_t$ (Li et al., 2022).

**Remark 1.** We remark that if $G(x(v,t)) = x(v,t)logx(v,t)$, then the gradient flow $\frac{\partial}{\partial t}x_t = -div(x_t\nabla\Phi_t)$ satisfies the heat equation $\frac{\partial}{\partial t}x_t = -\Delta x_t$, where $\Phi(v,t) = G'(v,t) = \log x(v,t)+1$. Mounting evidence shows that the message-passing mechanism in graph convolutional network can be formulated as a neural graph diffusion process Chamberlain et al. (2021).

This example indicates that the dynamic process of graph learning can be framed as a variational problem governed by a predefined gradient flow. Furthermore, it is possible to identify the most appropriate combination of energy function $\mathcal{E}(x_t)$ and gradient flow $\frac{\partial}{\partial t}x_t$ using machine learning technique, which sets the stage for a novel "meta-learning" paradigm. In what follows, we first unify existing GNN models into the reaction-diffusion model (RDM): $\frac{\partial x}{\partial t} = \Delta F(x) + R(x)$, where $F(\cdot)$ and $R(\cdot)$ are diffusion and reaction functions, respectively. By constraining the gradient flow to follow the characteristics of RDM, we further formulate it into a variational problem of mean-field control, where we seek to learn and integrate graph-specific control patterns into the gradient flow. Together, we present a physics-informed approach to jointly perform machine learning using the suitable relational inductive bias (aka. GNN learning mechanism) and fine-tune the GNN model instance through the Hamiltonian flows derived from the associated RDM.

## 2.2 Unifying Graph Neural Networks in Reaction Diffusion Model

In this section, we briefly review some representative GNNs and unify them in the umbrella of RDM. More details are shown in the Appendix A.

**GRAND** (Chamberlain et al., 2021). Graph neural diffusion (GRAND) draws inspiration from the heat diffusion equation, offering a unified mathematical framework for vanilla message-passing laws on graphs by:

$$\frac{\partial}{\partial t}x(t) = \text{div}[c\nabla x(t)] = c\Delta x(t), \tag{1}$$

where $F(x) = x$ and $R(x) = 0$. To simplify the problem formulation here, we assume $c$ is a homogeneous and time-invariant diffusion function. Thus, the intrinsic diffusion-reaction pattern can be further simplified as $\frac{\partial}{\partial t}x(t) = \Delta x(t)$ after dropping $c$.

**GraphBel** (Song et al., 2022). Extended from GRAND, the Beltrami diffusion on graph (GraphBel) proposed to use Beltrami flow to normalize the graph gradient as:

$$\frac{\partial}{\partial t}x(t) = \frac{1}{\|\nabla x\|}\text{div}\left(\frac{\nabla x(t)}{\|\nabla x(t)\|}\right), \tag{2}$$

where $\frac{\nabla x(t)}{\|\nabla x(t)\|}$ is a discrete analogue of the mean curvature operator. Without changing the diffusion-reaction property, we keep using $\Delta$ to indicate the normalized graph Laplacian operator here. Thus, Eq. 2 becomes $\frac{\partial}{\partial t}x(t) = \|\nabla x\|^{-1}\Delta x(t)$, where $\|\nabla x\| = \langle\nabla x,\nabla x\rangle^{\frac{1}{2}}$ is time-invariant magnitude of graph gradient. Since $\frac{1}{\|\nabla x\|}$ is decoupled with the divergence operator in Eq. 2, it is straightforward to derive $F(x) = x$ and $R(x) = 0$ in GraphBel.

**ACMP** (Wang et al., 2022). Inspired by interacting particle dynamics, Allen-Cahn message-passing (ACMP) graph neural network models both attractive and repulsive forces between two connected nodes during message-passing process using a predefined Allen-Cahn double-well potential function $f(x)$ (Allen & Cahn, 1979). The ACMP-based GNN models can be formulated as:

$$\frac{\partial}{\partial t}x(t) = m_x\left[\text{div}\left(\varepsilon_x^2\nabla x(t)\right) - f'(x)\right] \tag{3}$$

where $m_x$ is a hyper-parameter for the mobility and $\varepsilon_x$ a constant. Since $m_x$ and $\varepsilon_x$ do not determine the reaction-diffusion property, the PDE in ACMP can be simplified to $\frac{\partial}{\partial t}x(t) = \Delta x(t) - f'(x)$. In this scenario, $F(x) = x$ and $R(x) = -f'(x)$.

It is apparent that the characteristics of particular GNN instance are determined by a gradient flow $\frac{\partial x}{\partial t}$ associated with the reaction-diffusion model. As we explain next, gradient flow is a curve following the direction of steepest descent of a functional which essentially describes the working mechanism of the dynamical system. However, current GNN models simply apply the same PDE to all graph data, lacking a system-level understanding of the underlying learning mechanisms behind the PDE.

## 2.3 A Variational MFC Framework for Designing Novel GNN Models

Following the spirit of mean-field theory, we frame the dynamic process of learning graph feature representations as a complex system, where optimal features emerge as the system reaches *Nash-*

equilibrium governed by the control patterns learned from the underlying graph data. To achieve this, we present a variational framework of mean-field control to create the most suitable learning mechanism for GNN.

### 2.3.1 PROBLEM FORMULATION.

Suppose $\Psi(v)$ is situated in the continuous domain of a smooth positive density space. First, we construct a Lyapunov functional $\mathcal{E}(\Psi(v)) = \int_{\mathcal{G}} G(\Psi(v))dv$, where $G : \mathbb{R} \to \mathbb{R}$ is convex with $G''(\Psi) > 0$. If the gradient flow satisfies the RDM $\frac{\partial \Psi}{\partial t} = \Delta F(\Psi) + R(\Psi)$, minimizing $\mathcal{E}(\Psi(v))$ forms a dynamical system:

$$\frac{\partial}{\partial t}\mathcal{E}(\Psi_t) = -g(\Psi_t)^{-1}\frac{\partial}{\partial \Psi}\mathcal{E}(\Psi_t), \qquad g(\Psi) = \left(-\operatorname{div}(\frac{F'(\Psi)}{G''(\Psi)}\nabla) - \frac{R'(\Psi)}{G'(\Psi)}\right)^{-1}, \qquad (4)$$

where $g(\Psi)$ is the weighted elliptic operator (*Proof in Appendix Sec. A.2.*). To foreshadow the motivation for introducing the notion of control pattern in the MFC framework (in Section 2.3.3), we define the following two functionals to simplify the analytic expression of $g(\Psi)$.

**Definition 1. Transportation and reaction mobility functions.** Given $F$, $R$, and $G$, we define the transportation mobility function $\Theta_1(\Psi) = \frac{F'(\Psi)}{G''(\Psi)}$ and reaction mobility function $\Theta_2(\Psi) = -\frac{R(\Psi)}{G'(\Psi)}$.

The derivation of $g(\Psi)$ not only links objective functional $\mathcal{E}(\Psi)$ and RDM-based gradient flow, but also allows us to define the mean-field information metrics Ambrosio et al. (2005) as follows.

**Definition 2. Mean-field information metric.** Denote $\Phi$ as a smooth, positive density function. Given the elliptic operator $g$ (in Eq. 4), the metric between two densities $\Phi_1$ and $\Phi_2$ is:

$$\sigma(\Phi_1, \Phi_2) = \int_{\mathcal{G}} (\nabla \Phi_1, \nabla \Phi_2)\Theta_1 dv + \int_{\mathcal{G}} (\Phi_1, \Phi_2)\Theta_2 dv \qquad (5)$$

**Remark 2.** In the special case of $\Theta_1 = \Psi$ and $\Theta_2 = 0$ (first term in Eq. 6), the variational MFC problem is seeking the optimal transport $\psi_1$ to move the mass from $\Psi_0$ to $\Psi_1$ by minimizing $L_2$-*Wasserstein* metric. In another special case that $\Theta_1 = 0$ and $\Theta_2 = \Psi$ (second term in Eq. 6), the variational problem is corresponding to the Fisher-Rao metric, which has been well studied in information geometry (Amari, 2016). It is clear there are different choices of operator $g$ lead to different mean-field information metrics.

In the scenario of GNN, the input feature representations are often considered as $\Psi_0$ (initial state $t = 0$). The learned feature representations for down-stream task (such as node classification) are considered as $\Psi_1$ (terminal state $t = 1$). Given $\Psi_0$ and $\Psi_1$, a natural question is: *What is the most efficient way to transport $\Psi_0$ to $\Psi_1$?* The key to answering this optimal transport question is to study the critical point of the objective functional $\mathcal{E}(\Psi)$, which leads to the mean-field control problem.

**Definition 3. Mean-field control problem.** Consider a variational problem

$$\inf_{\psi_1, \psi_2, \Psi} \int_0^1 \left[\int_{\mathcal{G}} \frac{1}{2}\|\psi_1(v, t)\|^2 \Theta_1(\Psi(v, t)) + \frac{1}{2}|\psi_2(v, t)|^2 \Theta_2(\Psi(v, t))dv\right] dt, \qquad (6)$$

where the infimum is taken among all density functions $\Psi(v)$, vector fields $\psi_1$, and reaction rate functions $\psi_2$, such that

$$\partial_t \Psi(v, t) + \nabla \cdot (\Theta_1(\Psi(v, t))\psi_1(v, t)) = \psi_2(v, t)\Theta_2(\Psi(v, t)), \qquad (7)$$

with fixed initial and terminal density functions $\Psi_0, \Psi_1$.

### 2.3.2 GRAPH NEURAL NETWORK IS A MEAN-FIELD GAME.

We set up a mean-field game with $N$ players in a continuum of non-cooperative rational agents (graph nodes) distributed spatially in the graph $\mathcal{G}$ and temporally in $[0, 1]$. For an agent $v$ starting at $\Psi_0(v)$, the evolution of $\Psi(t, v)$ is completely determined by Eq. 7. To play the game over a time interval $[0, 1]$, each agent seeks to minimize the objective functional in Eq. 6, where the transportation and reaction mobility cost is incurred by each agent's own action. Following the pioneering work (Lasry & Lions, 2007), mean-field game is equivalent to variational formulation of MFC problem in **Definition 3**.

**Proposition 1. Hamiltonian flow in mean-field control problem**. Assume $\Psi(v, t) > 0$ for $t \in [0, 1]$. Then there exists a function $\Phi : [0, 1] \times \mathcal{G} \to \mathbb{R}$, such that the critical points of variational

problem Eq. 6 satisfy

$$\psi_1(v,t) = \nabla\Phi(v,t), \quad \psi_2(v,t) = \Phi(v,t) \tag{8}$$

with

$$\begin{cases} \partial_t \Psi(v,t) + \nabla \cdot (\Theta_1(\Psi(v,t))\nabla\Phi(v,t)) = \Theta_2(\Psi(v,t))\Phi(v,t), \\ \partial_t \Phi(v,t) + \frac{1}{2}\|\nabla\Phi(v,t)\|^2\Theta_1'(\Psi(v,t)) + \frac{1}{2}|\Phi(v,t)|^2\Theta_2'(\Psi(v,t)) = 0, \end{cases} \tag{9}$$

and $\Psi(v,0) = \Psi_0(v), \quad \Psi(v,t) = \Psi_1(v)$.

*Sketch of proof.* We introduce $\Phi$ as the Lagrange multiplier of variational problem (Eq. 6) constrained by the gradient flow in Eq. 7. Then we derive the solution of vector field $\psi_1$, reaction function $\psi_2$, and Hamiltonian flow in Eq. 9 by following the schema of saddle point problem (Benson et al., 2019). The detailed proof is shown in *Appendix Sec. A.3*.

**Remark 3.** If $\Theta_1 = \Psi$ and $\Theta_2 = 0$, the above formulation corresponds to the well-known Benamou-Brenier formula (Benamou & Brenier, 2000) in optimal transport. If $\Theta_1$ and $\Theta_2$ are positive functions then the objective functional in Eq. 6 is convex, making the derived gradient flow in Eq. 9 a minimizer of variational MFC problem (Li et al., 2022).

**Remark 4.** Suppose $\Psi_0$ is the initial graph representations. Given $\Theta_1$ and $\Theta_2$, Proposition 1 indicates that, at the mechanism level, the dynamical mechanics of feature representation learning from $\Psi_0$ to $\Psi_1$ is characterized by a Hamiltonian flow (Eq. 9), while at the model instance level, the alignment between the learned features $\Psi_1$ (terminal state) and the downstream task can be fine-tuned using a PDE-based GNN approach (Zhao et al., 2024) which is governed by the Hamiltonian flow.

### 2.3.3 DISCOVER MEAN-FIELD CONTROL PATTERNS $\Theta_1$ AND $\Theta_2$ FROM GRAPH DATA.

The variational framework for MFC problem provides a potential optimal solution for the objective functional in Eq. 6 by examining the saddle point. Like numerous entities in the universe operate, we propose that the information exchange mechanism in graph feature presentation learning (aka. relational inductive bias in GNN) also conforms to a dynamic system, adhering to the physical principles elucidated in Proposition 1. Particularly, the physical principle is characterized by the pre-selected transportation functional $\Theta_1$ and reaction functional $\Theta_2$. In contrast to the special cases of 2-Wasserstein distance (where $\Theta_1(\Psi) = \Psi, \Theta_2(\Psi) = 0$) and Fish-Rao metric (where $\Theta_1(\Psi) = 0$, $\Theta_2(\Psi) = \Psi$), $\Theta_1$ and $\Theta_2$ are essentially the weighted functions on each location $v$, acting as the expected **control patterns** that allow us to regulate the local message exchange during the evolution of graph representations $\Psi_t$. Naturally, we are motivated to learn the optimal control patterns $\Theta_1$ and $\Theta_2$, from the underlying graph data to improve the performance of GNN models.

In light of this, we present the following meta-learning paradigm that derives the most suitable learning mechanism from MFC problem and meanwhile optimizes model parameters using GNN backbones. By doing so, we expect to (1) enhance graph data learning performance on top of the existing GNN models and (2) establish an in-depth understanding of how individual node learns the best feature representations for themselves and the entire graph. Specifically, we introduce the functional Hamilton-Jacobi equations in positive density space (i.e., graph space) and define a Hamilton functional $\mathcal{H} : \mathcal{G} \times \mathcal{G} \to \mathbb{R}$ as follows

$$\mathcal{H}(\Psi, \Phi) = \int_{\mathcal{G}} \left( \frac{1}{2}\|\nabla\Phi\|^2\Theta_1(\Psi) + \frac{1}{2}|\Phi|^2\Theta_2(\Psi) \right) dv, \tag{10}$$

where the density function $\Psi$ serves as the state variable (akin to position), while the potential function $\Phi$ acts as the momentum variable in graph space.

### 2.4 GNN ABSTRACT LEARNING FRAMEWORK BY MEAN-FIELD CONTROL AND HAMILTONIAN MECHANICS

**MFC framework for GNNs.** In Sec. 2.2, we have shown the relationship between GNN model instance and reaction-diffusion equation. Despite many GNN models being fundamentally linked to the same PDE, they exhibit varied learning behaviors, yielding distinct learned feature representations. Within the variational framework of the MFC problem, such diversity can be attributed to the fact that different GNNs are driven by distinct objective functionals $\mathcal{E}(\Psi)$, each governed by unique physical principles. In Table 1, we summarize the energy variational functional $\mathcal{E}$, mobility functions $\Theta_1$ and $\Theta_2$, reaction-diffusion equation, and the corresponding Hamiltonian flows. Details can be found in *Sec. A.4 of the Appendix*.

It is clear that the objective functional $\mathcal{E}(\Psi) = \int_{\mathcal{G}} G(\Psi)dv$ (for crafting GNN mechanism) and the associated gradient flow in $\frac{\partial \Psi}{\partial t} = \Delta F(\Psi) + R(\Psi)$ (for optimizing GNN instance) are both related to transportation mobility function $\Theta_1$ and reaction mobility function $\Theta_2$. By capitalizing on this property, *Nash-GNN* emerges as the *first-ever* meta-learning graph learning approach. For clarity, we further summarize how GNN is formulated as mean-field games, as detailed in *Appendix A.5*.

**Network architecture of *Nash-GNN*.** Inspired by (Zhao et al., 2024), we propose an agnostic end-to-end deep model based on Hamiltonian mechanics, which characterizes information propagation in graph networks using a Hamiltonian-like structure. The overall network architecture is shown in Fig. 2. Specifically, we regard the potential energy $\Psi$ and latent function $\Phi$ ($\nabla\Phi$ is a flow vector field) in Eq. 9 as the position and momentum variables, respectively, in the Hamiltonian system, where the phase space $(\Psi, \Phi)$ characterizes the system's evolution (green arrow). Prior to $(\Psi(0), \Phi(0))$, we deploy a set of fully-connected layers $\mathcal{F}$ to project the observed nodal features $x$ to the energy function. There are two major inter-connected network components in *Nash-GNN*: (1) meta-learning component $\mathcal{M}$ (yellow box) for generating control patterns $\Theta_1$ and $\Theta_2$ based on the current estimation of phase space $(\Psi, \Phi)$ (indicated by blue arrow); and (2) PDE-based GNN instance $\mathcal{H}$ (gray box) for solving the evolution of Hamiltonian flow, where the terminal state of Hamiltonian flow is used to plug-in with the down-stream learning task (indicated by black arrow). The connection between $\mathcal{M}$ and $\mathcal{H}$ is the learned control patterns $\Theta_1$ and $\Theta_2$, as indicated by orange arrow in Fig. 2.

*Control pattern learning component $\mathcal{M}$: Generating $\Theta_1$ and $\Theta_2$ by input convex neural network (ICNN). Since energy function $\mathcal{H}(\Psi, \Phi)$ is completely deter-*

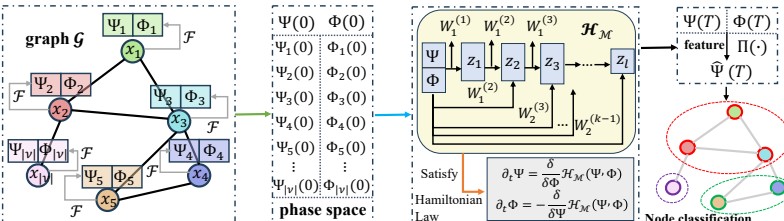

Figure 2: The network architecture of *Nash-GNN*.

mined by mobility function $\Theta_1$ and $\Theta_2$ (shown in Eq. 10), we propose to use a neural network $\mathcal{M}$ to establish the implicit mapping between the input $(\Psi, \Phi)$ and output $(\Theta_1, \Theta_2)$. As a crucial prerequisite for deriving the Hamiltonian flow outlined in Eq. 9, the objective function needs to be convex. Therefore, we use input convex neural network (Amos et al., 2017) as the backbone of $\mathcal{M}$, which yields the convex function instance in a recursive manner:

$$z^{(k+1)} = \sigma^{(k)}\left(W_1^{(k)}z^{(k)} + W_2^{(k)}(\Psi, \Phi) + b^{(k)}\right), \tag{11}$$

where $z^{(k)}$ denotes the output of $k^{th}$ layer. Each layer consists of two MLPs which project (1) the output from the previous layer $z^{k-1}$ (parameterized by $W_1^{(k)}$) and (2) the current phase-space $(\Psi, \Phi)$ (parameterized by $W_2^{(k)}$) and concatenate the output of two MLPs into $z^{(k+1)}$ by applying a non-linear activation $\sigma^{(k)}$ with a bias vector $b^{(k)}$. Thus, the output of meta-learning component $\mathcal{M}$ is the transportation function and reaction function, which allows to define the mobility measurements for each graph node based on the phase space $(\Psi, \Phi)$.

**GNN based on Hamiltonian flow.** In physics, systems evolve according to fundamental physical laws, with a (pre-defined) conserved quantity function $\mathcal{H}(\Psi, \Phi)$ that remains constant along the system's trajectory of evolution. This conserved quantity is commonly interpreted as the "system energy". We model the evolution of graph feature representations by following Hamiltonian equation:

$$\partial_t \Psi = \frac{\delta}{\delta \Phi}\mathcal{H}_{\mathcal{M}}(\Psi, \Phi), \quad \partial_t \Phi = -\frac{\delta}{\delta \Psi}\mathcal{H}_{\mathcal{M}}(\Psi, \Phi), \tag{12}$$

Table 1: Variational functionals $\mathcal{E}(\Psi) = \int_{\mathcal{G}} G(\Psi)dv$, diffusion function $F(\cdot)$, reaction function $R(\cdot)$, mobility functions $\Theta_1(\cdot)$ and $\Theta_2(\cdot)$, and Hamiltonian equations.

| Model | $\mathcal{E}(\Psi) = \int_{\mathcal{G}} G(\Psi)dv$ | $F(\Psi)$ | $R(\Psi)$ | $\Theta_1(\Psi) = \frac{F'}{G''}$ | $\Theta_2(\Psi) = -\frac{R}{G'}$ | Hamiltonian Equation |
|---|---|---|---|---|---|---|
| **GRAND** | $\int_{\mathcal{G}}(\Psi\log\Psi - 1)dv$ | $\Psi$ | $0$ | $\Psi$ | $0$ | $\begin{cases} \partial_t\Psi + \nabla\cdot(\Psi\nabla\Phi) = 0 \\ \partial_t\Phi + \frac{1}{2}\|\nabla\Phi\|^2 = 0 \end{cases}$ |
| **GraphBel** | $\int_{\mathcal{G}}(\frac{1}{2}\Psi^2)dv$ | $\Psi$ | $0$ | $1$ | $0$ | $\begin{cases} \partial_t\Psi + \nabla\cdot(\nabla\Phi) = 0 \\ \partial_t\Phi = 0 \end{cases}$ |
| **ACMP** | $\int_{\mathcal{G}} f(\Psi)dv$ | $\Psi$ | $-f'(\Psi)$ | $f''(\Psi)^{-1}$ | $1$ | $\begin{cases} \partial_t\Psi + \nabla\cdot(f''(\Psi)^{-1}\nabla\Phi) - \Phi = 0 \\ \partial_t\Phi - \frac{1}{2}\|\nabla\Phi\|^2\frac{f'''(\Psi)}{f''(\Psi)^2} = 0 \end{cases}$ |

with the initial features $(\Psi(0), \Phi(0))$ at time $t = 0$ being the vectors of potential energies. Supposing $\Psi(v) \in \mathbb{R}^d$ and $\Phi(v) \in \mathbb{R}^d$, we generate a $2d$-dimensional vectors that are then split into two equal halves: the first half serves as the feature (position) vector $\Psi$, while the second half represents the momentum vector guiding the system's evolution. Assuming a terminal time point $t = T$, the solution of the system is represented by $\Psi(T)$ and $\Phi(T)$, obtained through integration to derive the trajectory $(\Psi(t), \Phi(t))$ described in Eq. 12. After that, we apply the canonical projection function $\Pi$ to extract the concatenated feature vector $\Psi$ of the nodes from $(\Psi(T), \Phi(T))$, yielding $\Pi(\Psi(T), \Phi(T)) \rightarrow x(T)$, which can then be utilized for downstream tasks such as node classification. For clarify, we summarize the implemented details in Algorithm 1.

## 3 Experiments

**Experiment setup.** The evaluation on *Nash-GNN* not only includes benchmark with respect to existing state-of-the-art GNN models but also a *proof-of-concept* exploration to uncover novel insights into graph learning. Specifically, benchmark tests include (1) node classification and (2) graph classification. For **graph node classification**, we *first* apply our method to both heterophilic and homophilic datasets (sorted by homophily ratio $h$ (Zhu et al., 2020)): Texas ($h = 0.11$), Wisconsin ($h = 0.21$), Actor ($h = 0.22$), Squirrel ($h = 0.22$), Chameleon ($h = 0.23$), Cornell ($h = 0.3$), Citeseer ($h = 0.74$), Pubmed ($h = 0.8$) and Cora ($h = 0.81$), where $h$ indicate the fraction of edges that connect nodes with the same label. We *then* verify the performance of *Nash-GNN* on large-scale dataset from OGB (Open Graph Benchmark) (Hu et al., 2020), i.e., ogbn-arxiv and ogbn-products. For **graph classification**, we *first* conduct an experiment on the benchmark results on TUDataset (Morris et al., 2020) including MUTAG, NCI1, ENZYMES, D&D, PTC_FM, IMDB-B and PROTEINS. To demonstrate the generality and scalability of our proposed model, we *then* apply the *Nash-GNN* to human connectomes for disease diagnosis, we use the processed neuroimaging data in the published datasets (Xu et al., 2024): ABIDE (Autism), ADNI (Alzheimer) (Weiner et al., 2010), OASIS (Alzheimer) (LaMontagne et al., 2019), PPMI (Parkinson), where we use regional BOLD (blood oxygenation level-dependent) time series as the graph embedding and functional connectivity (FC) with ALL atlas (116 regions) (Tzourio-Mazoyer et al., 2002) as the adjacency matrix. The data description is shown in *Sec. B.1*.

We compare the performance with various benchmark GNN models, including vanilla GCN (Kipf & Welling, 2017), GAT (Veličković et al., 2018), GraphSAGE (Hamilton et al., 2017b), GraphCON (Rusch et al., 2022), GraphBel (Song et al., 2022), GRAND (Chamberlain et al., 2021), ACMP (Wang et al., 2022) HANG (Zhao et al., 2024), GIN (Xu et al., 2018) and AM-GCN (Wang et al., 2020). For conventional graph data in node classification experiments, we follow a challenging data-splitting method published in (Zheng et al., 2021) (graph robustness benchmark), with $60\%$ for training, $10\%$ for validation, and the rest of the nodes for the testing set. We adhere to all standard OGB evaluation settings. For TUDataset, we report the 10-fold cross-validation (follow (Ranjan et al., 2020)) results on different models. For human connectome data, we report the 5-fold cross-validation results.

### 3.1 Benchmark Evaluations: GNN Is A Mean-Field Game

**Performance on graph node classification.** *Results.* Table 2 and Table 8 list the comparison results for nine classic graph datasets and two large-scale datasets on nine methods. *Nash-GNN* achieves SOTA performance on heterophilic and homophilic as well as large-scale graph data over the existing hand-designed GNN models. Moreover, we perform two ablation studies in terms of different Hamiltonian energy functions and PDE solvers, the results are shown in *Appendix B.3*.

*Discussion.* These results provide strong evidence that our variational framework is able to customize the most suitable mobility functions for different graph data, which contributes to enhanced learning performance compared to other "one-size-fits-fall" approaches. More importantly, this experiment supports our vision that "*GNN is a mean-field game*".

**Performance on graph classification.** In graph classification experiments, we include a "global_max_pool" function and a fully connected layer to achieve the graph classification task. *Results on TUDataset.* Table 3 presents the benchmark results for six classic methods on the popular TUDataset. Our *Nash-GNN*, demonstrates strong performance across various types of graph data, including molecules, bioinformatics, and social networks, outperforming several existing hand-designed GNN models.

Table 2: Top: Test accuracies (%) on nine graph networks for node classification task. Statistical significance is assessed based on 20 resampling tests conducted using a randomized seed. '∗' means statistically significance with $p \leq 0.05$. Bottom: Diagnosis accuracies (%) on disease-based datasets.

| Dataset | GCN | GAT | GraphSAGE | GraphCON | GraphBel | GRAND | ACMP | HANG | *Nash-GNN* |
|---|---|---|---|---|---|---|---|---|---|
| **Texas** | $61.48_{\pm7.68}$ | $61.48_{\pm5.61}$ | $81.85_{\pm5.46}$ | $84.20_{\pm3.01}$ | $85.71_{\pm2.96}$ | $85.52_{\pm0.57}$ | $86.00_{\pm3.21}$ | $85.56_{\pm1.54}$ | $\mathbf{94.00^*_{\pm2.21}}$ |
| **Wisconsin** | $55.00_{\pm5.79}$ | $49.86_{\pm3.47}$ | $81.07_{\pm3.82}$ | $87.93_{\pm4.30}$ | $86.95_{\pm3.53}$ | $87.35_{\pm5.37}$ | $86.49_{\pm4.30}$ | $84.27_{\pm4.75}$ | $\mathbf{88.22^*_{\pm4.16}}$ |
| **Actor** | $29.05_{\pm1.69}$ | $26.34_{\pm0.87}$ | $32.72_{\pm3.62}$ | $35.62_{\pm1.54}$ | $33.58_{\pm1.61}$ | $33.20_{\pm1.25}$ | $33.65_{\pm1.95}$ | $35.12_{\pm1.06}$ | $\mathbf{36.25^*_{\pm1.38}}$ |
| **Squirrel** | $39.27_{\pm1.09}$ | $36.05_{\pm1.69}$ | $39.76_{\pm1.88}$ | $34.90_{\pm1.90}$ | $39.08_{\pm6.49}$ | $35.46_{\pm1.01}$ | $36.56_{\pm1.52}$ | $39.32_{\pm1.28}$ | $\mathbf{40.38^*_{\pm0.60}}$ |
| **Chameleon** | $55.65_{\pm3.54}$ | $51.57_{\pm4.93}$ | $55.12_{\pm1.84}$ | $48.31_{\pm2.77}$ | $46.79_{\pm1.61}$ | $46.03_{\pm1.89}$ | $46.85_{\pm1.95}$ | $59.85_{\pm2.12}$ | $\mathbf{60.14_{\pm0.22}}$ |
| **Cornell** | $50.00_{\pm7.25}$ | $40.37_{\pm4.61}$ | $67.78_{\pm7.81}$ | $82.16_{\pm6.17}$ | $78.53_{\pm1.28}$ | $80.00_{\pm2.31}$ | $79.73_{\pm2.86}$ | $77.41_{\pm4.01}$ | $\mathbf{83.59^*_{\pm2.23}}$ |
| **Citeseer** | $66.51_{\pm12.75}$ | $46.13_{\pm19.72}$ | $63.16_{\pm9.34}$ | $74.84_{\pm0.49}$ | $69.62_{\pm0.56}$ | $74.98_{\pm1.45}$ | $75.07_{\pm2.17}$ | $73.75_{\pm1.80}$ | $\mathbf{77.61^*_{\pm1.38}}$ |
| **Pubmed** | $88.63_{\pm1.31}$ | $87.36_{\pm1.34}$ | $88.71_{\pm0.34}$ | $88.78_{\pm0.46}$ | $86.97_{\pm0.37}$ | $88.44_{\pm0.34}$ | $87.76_{\pm1.24}$ | $89.93_{\pm0.27}$ | $\mathbf{90.75_{\pm0.27}}$ |
| **Cora** | $86.86_{\pm0.63}$ | $87.04_{\pm0.69}$ | $81.38_{\pm6.24}$ | $86.27_{\pm0.51}$ | $82.60_{\pm0.64}$ | $87.53_{\pm0.59}$ | $82.91_{\pm2.62}$ | $84.56_{\pm1.21}$ | $\mathbf{87.80_{\pm0.47}}$ |

Table 3: Performace on TUDataset.

| Dataset | MUTAG | NCI1 | ENZYMES | D&D | PTC_FM | IMDB-B | PROTEINS |
|---|---|---|---|---|---|---|---|
| GCN | $0.730_{\pm0.022}$ | $0.609_{\pm0.020}$ | $0.247_{\pm0.010}$ | $0.705_{\pm0.010}$ | $0.608_{\pm0.022}$ | $0.740_{\pm0.002}$ | $0.706_{\pm0.008}$ |
| GAT | $0.727_{\pm0.021}$ | $0.574_{\pm0.026}$ | $0.265_{\pm0.017}$ | $0.693_{\pm0.012}$ | $0.609_{\pm0.021}$ | $0.727_{\pm0.012}$ | $0.705_{\pm0.005}$ |
| GraphSAGE | $0.732_{\pm0.023}$ | $0.705_{\pm0.003}$ | $0.300_{\pm0.014}$ | $0.715_{\pm0.008}$ | $0.602_{\pm0.019}$ | $0.729_{\pm0.010}$ | $0.704_{\pm0.005}$ |
| GCNII | $0.728_{\pm0.022}$ | $0.691_{\pm0.003}$ | $0.444_{\pm0.022}$ | $0.706_{\pm0.008}$ | $0.616_{\pm0.013}$ | $0.694_{\pm0.007}$ | $0.695_{\pm0.010}$ |
| AM-GCN | $0.803_{\pm0.015}$ | $0.665_{\pm0.002}$ | $0.411_{\pm0.022}$ | $0.741_{\pm0.004}$ | $0.620_{\pm0.004}$ | $0.505_{\pm0.017}$ | $0.713_{\pm0.006}$ |
| GIN | $0.814_{\pm0.015}$ | $0.750_{\pm0.140}$ | $0.496_{\pm0.045}$ | $0.730_{\pm0.033}$ | $0.590_{\pm0.033}$ | $0.728_{\pm0.009}$ | $0.715_{\pm0.017}$ |
| *Nash-GNN* | $\mathbf{0.834^*_{\pm0.019}}$ | $\mathbf{0.749^*_{\pm0.004}}$ | $\mathbf{0.546^*_{\pm0.015}}$ | $\mathbf{0.770^*_{\pm0.009}}$ | $\mathbf{0.644^*_{\pm0.003}}$ | $\mathbf{0.751^*_{\pm0.013}}$ | $\mathbf{0.740^*_{\pm0.005}}$ |

Table 4: Diagnosis accuracies (%) on disease-based datasets.

| Dataset | GCN | GAT | GraphSAGE | GraphCON | GraphBel | GRAND | ACMP | HANG | *Nash-GNN* |
|---|---|---|---|---|---|---|---|---|---|
| **ABIDE** | $54.85_{\pm0.56}$ | $54.29_{\pm0.26}$ | $54.41_{\pm0.56}$ | $54.51_{\pm0.31}$ | $54.72_{\pm0.51}$ | $\mathbf{62.79_{\pm12.79}}$ | $54.85_{\pm0.64}$ | $51.92_{\pm0.97}$ | $62.78^*_{\pm1.26}$ |
| **ADNI** | $76.62_{\pm3.76}$ | $72.36_{\pm3.41}$ | $75.10_{\pm3.4}$ | $71.24_{\pm2.05}$ | $91.75_{\pm8.64}$ | $78.10_{\pm16.51}$ | $92.68_{\pm7.68}$ | $91.47_{\pm4.17}$ | $\mathbf{95.82^*_{\pm1.50}}$ |
| **OASIS** | $85.95_{\pm0.05}$ | $86.06_{\pm0.22}$ | $86.01_{\pm0.23}$ | $86.23_{\pm0.27}$ | $85.93_{\pm0.25}$ | $85.54_{\pm3.10}$ | $85.79_{\pm0.23}$ | $85.89_{\pm0.35}$ | $\mathbf{89.22^*_{\pm0.18}}$ |
| **PPMI** | $68.02_{\pm11.57}$ | $64.96_{\pm7.33}$ | $68.02_{\pm11.14}$ | $64.54_{\pm9.31}$ | $71.86_{\pm4.05}$ | $71.43_{\pm1.85}$ | $\mathbf{72.45_{\pm2.21}}$ | $70.91_{\pm2.24}$ | $\mathbf{74.03_{\pm3.11}}$ |

*Discussion.* These results provide compelling evidence that our variational framework is well-suited for various types of graph datasets, resulting in improved learning performance compared to approaches tailored to specific datasets.

*Results on human connectomes.* Table 4 summarizes the diagnostic performance across six disease-based datasets, where we predict the likelihood of developing neurological disease in unseen subjects using graph data. The experimental findings demonstrate that our method exhibits significant effectiveness in disease diagnosis, suggesting the promising clinical value of deploying our approach in disease early diagnosis.

*Discussion.* Atypical neuron growth/loss is the hallmark of many neurological diseases (Dickson, 2010; Lord et al., 2018). Meanwhile, the prion-like mechanism (i.e., misfolded proteins spread like an infection in the brain) has been frequently reported in many neuroscience studies (Frost & Diamond, 2016; Guo & Lee, 2014), where network topology plays a vital role in determining the kinetic of pathology propagation (Palop et al., 2006). In addition to the standard attention mechanism in GNN (Veličković et al., 2018), the MFC framework in *Nash-GNN* allows us to uncover the dynamic mechanism of disease progress from a system perspective, as shown below.

## 3.2 METHOD EXPLORATION: NEW INSIGHT OF GRAPH LEARNING BEYOND ATTENTION

In this section, we put the spotlight on the transport mobility function $\Theta_1$ since this mobility function is intuitively relevant to the message-exchanging mechanism in GNNs. For each graph dataset, one of the outputs of *Nash-GNN* is the learned $\Theta_1$ at each graph node, where we essentially employ ICNN backbone (Eq. 11 in the abstract learning module $\mathcal{M}$) to generate a convex function based on the flow information $\Psi$. Assuming the latent convex function is a polynomial function, we compute the mean polynomial power $\alpha$ at each graph node by applying uni-variate polynomial fitting for each element and then averaging the degrees of polynomial power. After that, we conduct several post-hoc analyses at graph level and node level, respectively. *First*, we use the averaged polynomial power (across nodes) to express the graph homophily ratio $h$, to uncover the new insight into how the dynamics of information exchange in GNN correlates with the properties of the graph data. *Second*, we extend this global analysis to each graph node with the hypothesis that mobility of spreading node

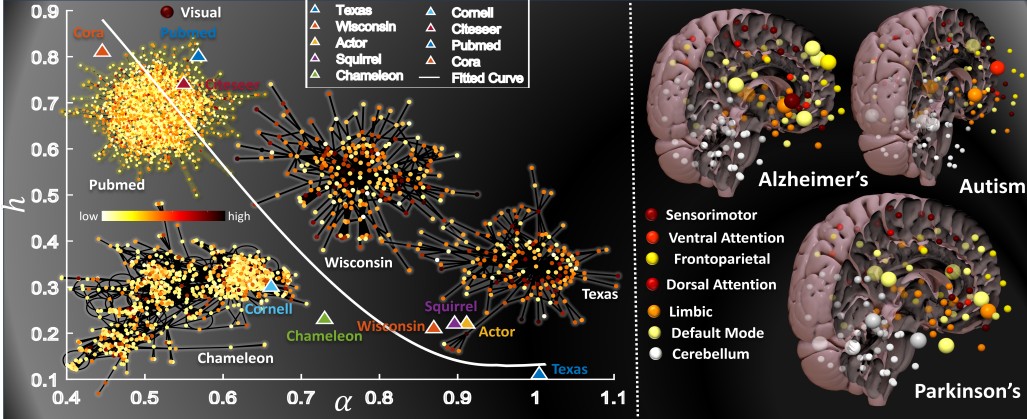

Figure 3: Left: Correlation between graph homophily ratio $h$ (y-axis) and the learned control pattern $\Theta_1(\Psi) = \Psi^\alpha$ (x-axis). $h$ is a global measurement where a large degree indicates a better alignment between graph topology and the consistency of labels across connected nodes. We use a polynomial function to fit $\Theta_1$ for each node, where the degree of $\alpha$ is inversely proportional to the freedom of local information exchange. It is clear that the learned control pattern is highly correlated with the heuristic measurement. Right: Top ten significant brain regions associated with the pathophysiological mechanism of AD, Autism, and PD.

embeddings (related to neuropathology burdens) underlines the biological mechanism in disease progression.

*Results.* As the white curve shown in Fig. 3.2 left, the $\alpha \sim h$ relationship across nine graph dataset indicates a notable anti-correlation. It implies that the effective way to perform graph learning is to promote information exchange on homophilic graphs (such as Cora and Pubmed) while constraining the diffusion of information between connected nodes with different labels in heterophilic graphs (such as Texas and Wisconsin). The reason behind is rooted in the MFC objective functional (Eq. 6 that larger degree of $\Theta_1$ encourages the optimization process favoring smaller flows $\Psi_1$ which is aligned with the heuristic of penalizing information exchange in heterophilic graphs. Furthermore, we display the learned $\Theta_1$ at node level for Texas, Wisconsin, Chameleon, and Pubmed in Fig. 3.2 left, where bright yellow and dark red denote for small and large degree of $\Theta_1(\Psi(v))$, respectively.

*Discussion.* Similarly, we conduct the same post-hoc analysis to investigate the biological underpinning between the learned node-wise transport mobility degree and pathophysiological mechanism of disease progression. In Fig. 3.2 right, we use large size node to indicate the larger mobility of the underlying node (associated with smaller degree of $\Theta_1(\Psi(v))$). It is interesting to find that the brain regions with high dynamics for pathology propagation are closely associated with our current findings on disease etiology. Take Alzheimer's disease (AD) for example, resting-state fMRI studies have identified significant alterations in BOLD signal dynamics within the default mode network (DMN), which may indicate abnormalities in functional connectivity (Varma et al., 2017). Here, we use machine learning techniques to provide another piece of data-driven evidence to support this finding as most of the large-size nodes are located in DMN. Additionally, our findings reveal that (1) increased mobility of pathological factors in the cerebellum correlates with the progression of Parkinson's disease, and (2) accelerated neuron overgrowth in the dorsal attention and limbic networks, as well as the cerebellum, may potentially be the contributing factor to autism. These promising results underscore the new window to answer neuroscience questions using explainable machine learning techniques.

## 4 CONCLUSIONS

In this work, we embarked on a new abstract learning framework for GNN to customize graph neural networks for various graph-based machine learning tasks. We integrate the theory of mean-field control into GNNs to enhance our understanding and guide the development of deep models for new graph datasets. We also provide an end-to-end solution using Hamiltonian flows to jointly learn suitable inductive bias for GNN model and fit the customized GNN model to the underlying graph data. Our approach is thoroughly evaluated on standard benchmark datasets, and we explore fundamental principles of graph learning as an interactive dynamical system, which not only advances GNN understanding but also contributes to the broader field of graph-based machine learning.

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

## A  PROOF OF PROPOSITIONS AND DETAILED FORMULATIONS AND EXPLANATIONS

### A.1  THE EXPLANATION OF DEFINITION 1

The motivation of formulating $\Theta_1 = \frac{F'(\Psi)}{G''(\Psi)}$ and $\Theta_2 = -\frac{R(\Psi)}{G'(\Psi)}$ is to facilitate (1) linking the diffusion term $F(\Psi)$ and reaction term $R(\Psi)$ in RDM to energy functional $\mathcal{E}(\Psi) = \int_{\mathcal{G}} G(\Psi(v))dv$ and (2) generalizing existing GNNs into the RDM framework. As described in Sec. 2.2, current PDE-based GNN models can be regarded as the RDM with empirically defined diffusion functional instance $F(\Psi)$ and reaction functional instance $R(\Psi)$. In our approach, the behavior of diffusion $F$ and reaction $R$ is not fixed for all graph data. Instead, we learn the most appropriate diffusion and reaction functions through the transport functional $\Theta_1$ and reaction mobility functional $\Theta_2$.

### A.2  PROOF OF PROPOSITION 1

To prove Proposition 1, we consider constructing a Lyapunov functional $\mathcal{E} : \mathcal{G} \to \mathbb{R}$ to study the RDM, thus considering $\mathcal{E}(\Psi) = \int G(\Psi(v))dv$, where $G : \mathbb{R} \to \mathbb{R}$ is a convex function with $G''(\Psi) > 0$. In such cases, we have

$$\frac{d}{dt}\mathcal{E}(\Psi(t,\cdot)) = \int G'(\Psi(t,v)) \cdot \partial_t \Psi(t,v)dv = \int G'(\Psi(t,v))(\Delta F(\Psi(t,v)) + R(\Psi(t,v)))dv$$

$$= -\int (\nabla G'(\Psi(t,v)), \nabla G'(\Psi(t,v))) \frac{F'(u(t,v))}{G''(\Psi(t,v))}dv + \int G'(\Psi(t,v))^2 \frac{R(\Psi(t,v))}{G'(\Psi(t,v))}dv$$

$$(13)$$

where we apply $\nabla G'(\Psi) = G''(\Psi)\nabla\Psi$ in Eq. 13. Herein, we assume $-\frac{R}{G'} > 0$ and $F'(\Psi) > 0$ for $\Psi > 0$, thus we have $\frac{d}{dt}\mathcal{E}(\Psi) \leq 0$, indicting that functional $\mathcal{E}(\Psi)$ is not increasing along flow. The decay behavior described above suggests a gradient flow formulation for the dynamics outlined in RDM. To refine and clarify this concept, we introduce an inverse of the weighted elliptic operator

$$g(\Psi) := \left(-\nabla \cdot \left(\frac{F'(\Psi)}{G''(\Psi)}\nabla\right) - \frac{R(\Psi)}{G'(\Psi)}\right)^{-1} \tag{14}$$

Thus we have

$$\partial_t \Psi = -g(\Psi)^{-1}\frac{\delta}{\delta\Psi}\mathcal{G}(\Psi) = -\left(-\nabla \cdot \left(\frac{F'(\Psi)}{G''(\Psi)}\nabla\right) - \frac{R(\Psi)}{G'(\Psi)}\right)\frac{\delta}{\delta\Psi}\mathcal{G}(\Psi)$$

$$= \nabla \cdot \left(\frac{F'(\Psi)}{G''(\Psi)}\nabla G'(\Psi)\right) + \frac{R(\Psi)}{G'(\Psi)}G'(\Psi) = \Delta F(\Psi) + R(\Psi) \tag{15}$$

where $\frac{\delta}{\delta\Psi}$ denotes the $L^2$ first variation w.r.t. $\Psi \in \mathcal{M}(\mathcal{E})$. Based on the above notation, the dissipation of Lyapunov functional $\mathcal{E}$ along RDM satisfies $\frac{d}{dt}\mathcal{E}(\Psi) = -\int \left(\frac{\delta}{\delta\Psi}\mathcal{G}(\Psi), g(\Psi)^{-1}\frac{\delta}{\delta\Psi}\mathcal{G}(\Psi)\right) dv \leq 0$.

### A.3  PROOF OF PROPOSITION 2

To prove Proposition 2, we first rewrite the variables in variational problem Eq. 6 of the main text as

$$q_1(t,v) = \Theta_1(\Psi)\psi_1(t,v), \quad q_2(t,v) = \Theta_2(\Psi)\psi_2(t,v), \tag{16}$$

Thus the variational problem Eq. 6 forms

$$\inf_{m_1,m_2,u}\left\{\int_0^1 \int_{\mathcal{G}} \frac{\|q_1(t,v)\|^2}{2\Theta_1(\Psi(t,v))} + \frac{|q_2(t,v)|^2}{2\Theta_2(\Psi(t,v))}dvdt : \right.$$

$$\left. \partial_t\Psi(t,v) + \nabla \cdot q_1(t,v) = q_2(t,v), \quad \text{fixed } \Psi_0, \Psi_1\right\}. \tag{17}$$

Denote the Lagrange multiplier of Eq. 17 by $\Phi$. We consider the following saddle point problem

$$\inf_{q_1, q_2, \Psi} \sup_{\Phi} \mathcal{L}\left(q_1, q_2, \Psi, \Phi\right), \tag{18}$$

with

$$\mathcal{L}\left(q_1, q_2, \Psi, \Phi\right) = \int_0^1 \int_{\mathcal{G}} \left\{ \frac{\|q_1(t,v)\|^2}{2\Theta_1(\Psi(t,v))} + \frac{|q_2(t,v)|^2}{2\Theta_2(\Psi(t,v))} \right.$$
$$\left. + \Phi(t,v)\left(\partial_t \Psi(t,v) + \nabla \cdot q_1(t,v) - q_2(t,v)\right)\right\} dvdt. \tag{19}$$

By finding the saddle point of $\mathcal{L}$, we have

$$\begin{cases} \frac{\delta}{\delta q_1}\mathcal{L} = 0, \\ \frac{\delta}{\delta q_2}\mathcal{L} = 0, \\ \frac{\delta}{\delta \Psi}\mathcal{L} = 0, \\ \frac{\delta}{\delta \Phi}\mathcal{L} = 0, \end{cases} \Rightarrow \begin{cases} \frac{q_1}{\Theta_1} = \nabla\Phi, \\ \frac{q_2}{\Theta_2} = \Phi, \\ -\frac{1}{2}\frac{\|q_1\|^2}{\theta_1^2}\Theta_1' - \frac{1}{2}\frac{|q_2|^2}{\Theta_2^2}\Theta_2' - \partial_t\Phi = 0, \\ \partial_t\Psi + \nabla \cdot q_1 - q_2 = 0, \end{cases} \tag{20}$$

where $\frac{\delta}{\delta q_1}, \frac{\delta}{\delta q_2}, \frac{\delta}{\delta \Psi}, \frac{\delta}{\delta \Phi}$ are $L^2$ first variations w.r.t functions $q_1, q_2, \Psi, \Phi$, respectively. After that, by substituting the above two row equations into the last two row equations of Eq. 20, we derive the PDE pair Eq. 9 in the main text.

### A.4 LYAPUNOV FUNCTINALS, REACTION-DIFFUSION EQUATIONS AND HAMILTONIAN EQUATIONS

**For GRAND case**, let $G(\Psi) = \Psi \log \Psi - 1$, $F(\Psi) = \Psi$, $R(\Psi) = 0$, thus we have based on Definition 1

$$\Theta_1(\Psi) = \frac{F'(\Psi)}{G''(\Psi)} = \Psi, \quad \Theta_2(\Psi) = -\frac{R(\Psi)}{G'(\Psi)} = 0 \tag{21}$$

According to Eq. 14, the metric forms

$$g(\Psi)(\sigma_1, \sigma_2) = \int_{\mathcal{G}} \left(\nabla\Phi_1(v), \nabla\Phi_2(v)\right) \Psi(v)dv. \tag{22}$$

Consider the relations $\sigma_i = -\nabla \cdot (\Psi\nabla\Phi_i)$ for $i = 1, 2$. In this scenario, the mean-field information metric aligns with the *Wasserstein*-2 metric, as discussed in (Ambrosio et al., 2005). The gradient flow of $\mathcal{E}(\Psi)$, denoted as the negative Boltzmann-Shannon entropy, on the graph space $(\mathcal{G}, g)$, corresponds to the heat equation described by Eq. 1 in the main text, i.e.,

$$\partial_t \Psi = \nabla \cdot (\Psi\nabla G'(\Psi)) = \nabla \cdot (\Psi G''(\Psi)\nabla\Psi) = \Delta\Psi, \tag{23}$$

which is equivalent to Eq. 1 in the main text. The dissipation of $\mathcal{E}(\Psi)$ forms $\int_{\mathcal{G}} \|\nabla \log \Psi(v)\|^2 \Psi(v)dv$. And the Hamilton-Jacobi equation in $(\mathcal{G}, g)$ follows $\partial_t\mathcal{U}(t, \Psi) + \frac{1}{2}\int_{\mathcal{G}} \left\|\nabla\frac{\delta}{\delta\Psi(v)}\mathcal{U}(t, \Psi)\right\|^2 \Psi(v)dv = 0$, where $\frac{\delta}{\delta\Psi(v)}\mathcal{U}(t, \Psi) = \Phi(t, v)$. According to Eq. 10 in the main text, its "characteristics" in graph space $(\mathcal{G}, g)$ satisfy

$$\begin{cases} \partial_t\Psi + \nabla \cdot (\Psi\nabla\Phi) = 0, \\ \partial_t\Phi + \frac{1}{2}\|\nabla\Phi\|^2 = 0. \end{cases} \tag{24}$$

**For GraphBel case**, let $G(\Psi) = \frac{1}{2}\Psi^2$, $F(\Psi) = \Psi$, $R(\Psi) = 0$, thus we have

$$\Theta_1(\Psi) = \frac{F'(\Psi)}{G''(\Psi)} = 1, \quad \Theta_2(\Psi) = -\frac{R(\Psi)}{G'(\Psi)} = 0 \tag{25}$$

The metric forms

$$g(\Psi)(\sigma_1, \sigma_2) = \int_{\mathcal{G}} \left(\nabla\Phi_1(v), \nabla\Phi_2(v)\right) dv \tag{26}$$

with $\sigma_i = -\nabla \cdot (\nabla\Phi_i)$, $i = 1, 2$. The gradient flow of $\mathcal{E}$ in $(\mathcal{G}, g)$ forms

$$\partial_t\Psi = \nabla \cdot (\nabla G'(\Psi)) = \nabla \cdot (G''(\Psi)\nabla\Psi) = (\nabla\Psi)^{-1}\Delta\Psi, \tag{27}$$

which is equivalent to Eq. 2 in the main text. The dissipation of $\mathcal{E}(\Psi)$ forms $\int_{\mathcal{G}} \|\nabla\Psi))\|^2 dv$. And the Hamilton-Jacobi equation in $(\mathcal{G}, g)$ follows $\partial_t \mathcal{U}(t, \Psi) + \frac{1}{2}\int_{\mathcal{G}} \left\|\nabla \frac{\delta}{\delta\Psi(v)}\mathcal{U}(t, \Psi)\right\|^2 dv = 0$. Its "characteristics" in graph space $(\mathcal{G}, g)$ satisfy

$$\begin{cases} \partial_t \Psi + \nabla \cdot (\nabla\Phi) = 0, \\ \partial_t \Phi = 0. \end{cases} \tag{28}$$

**For ACMP case**, let $f \in C^2(\mathbb{R})$ be a given function. Consider $F(\Psi) = \Psi$, $G(\Psi) = f(\Psi)$, $R(\Psi) = -f'(\Psi)$. We have

$$\Theta_1(\Psi) = \frac{F'(\Psi)}{G''(\Psi)} = f''(\Psi)^{-1}, \quad \Theta_2(\Psi) = -\frac{R(\Psi)}{G'(\Psi)} = 1. \tag{29}$$

The metric forms

$$g(\Psi)(\sigma_1, \sigma_2) = \int_{\mathcal{G}} (\nabla\Phi_1, \nabla\Phi_2) f''(\Psi)^{-1} dv + \int_{\mathcal{G}} \Phi_1\Phi_2 dv, \tag{30}$$

with $\sigma_i = -\nabla \cdot (f''(\Psi)^{-1}\nabla\Phi_i) + \Phi_i, i = 1, 2$. The gradient flow of $\mathcal{E}$ in graph space $(\mathcal{G}, g)$ satisfy

$$\partial_t \Psi(t, v) = \Delta\Psi(t, v) - f'(\Psi(t, v)), \tag{31}$$

which is equivalent to Eq. 3 in the main text. And the dissipation of $\mathcal{E}(\Psi)$ satisfies $\int_{\mathcal{G}} \|\nabla f'(\Psi)\|^2 f''(\Psi)^{-1} dv + \int_{\mathcal{G}} |f'(\Psi)|^2 dv$. And the Hamilton-Jacobi equation in $(\mathcal{G}, g)$ follows $\partial_t \mathcal{U}(t, \Psi) + \frac{1}{2}\int_{\mathcal{G}} \left\|\nabla \frac{\delta}{\delta\Psi(v)}\mathcal{U}(t, \Psi)\right\|^2 f''(\Psi(v))^{-1} dv + \frac{1}{2}\int_{\mathcal{G}} \left|\frac{\delta}{\delta\Psi(v)}\mathcal{U}(t, \Psi)\right|^2 dv = 0$. Thus, its "characteristics" in graph space $(\mathcal{G}, g)$ satisfy

$$\begin{cases} \partial_t \Psi + \nabla \cdot (f''(\Psi)^{-1}\nabla\Phi) - \Phi = 0, \\ \partial_t \Phi - \frac{1}{2}\|\nabla\Phi\|^2 \frac{f'''(\Psi)}{f''(\Psi)^2} = 0. \end{cases} \tag{32}$$

### A.5 GNN Is A Mean Filed Game

In the following, we emphasize the explanation of the principle of how GNN is formulated as a mean-field game, the evolution of *Nash-GNN* from GNN is summarized in Fig. 4.

GNN is a dynamical system. Simply put, GNN is a black box that converts the initial feature representations into a latent subspace by a set of information exchanges (constrained by graph topology) and projection (using a mapping function shared by all graph nodes). As GNNs often consist of multiple layers, the evolution of feature representation from the initial state (input graph embeddings) to the terminal state (last layer of GNN) can be regarded as a time-dependent dynamical system, where the dynamics is determined by a

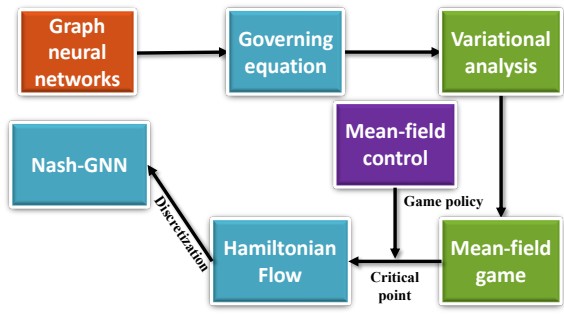

Figure 4: The evolution of *Nash-GNN* from GNN.

governing equation (in the form of PDE). In the reminiscent of the Brachistochrone problem [1] (a classic physics problem that involves finding the curve down which a bead sliding under the influence of gravity will travel in the least amount of time between two points), the powerful calculus of variations (COV) allows us to generate various governing equations, providing a necessary condition that a function must satisfy in order to be an extremum of a given functional.

GNN-PDE-COV interplay. Inspired by recent PDE-based deep models such as Neural ODE and GRAND, we frame the layer-by-layer feed-forward process $x^{(l+1)} = \sigma(AWx^{(l)})$ as a dynamical system, where the time-evolving mechanics is determined by the graph heat equation $\frac{\partial x(t)}{\partial t} =$

---

[1]https://en.wikipedia.org/wiki/Brachistochrone_curve

$\Delta x(t)$. Here, $A$ and $\Delta$ denote the normalized adjacency matrix and Laplacian matrix, $W$ is the learnable mapping parameters, and $\sigma$ denotes the nonlinear activation function. Indeed, the evolution of the heat equation forms the gradient flow of the Dirichlet energy $\varepsilon(x) = \frac{1}{2}\int |x|^2$. Thus, we have established a connection between the GNN model instance in the discrete domain and the equivalent variation functional in the continuous domain, where the governing equation is acting as a stepping stone.

Setup of mean-field game for GNN. Following the notion of mean-field game, each graph node is acting as an agent. The game is to find the best feature presentations for all graph nodes that minimize the loss function in GNN. In a mean field game, each agent (aka. graph node) makes decisions based on both their individual state and the aggregate effect of the states and actions of all other agents, often referred to as the "mean field." The primary goal is to find a Nash equilibrium, which is a strategic decision-making in very large populations of interacting agents such that no agent can benefit by changing their strategy while others keep theirs unchanged. Mathematically, mean field games often involve solving coupled partial differential equations such as Hamilton-Jacobi-Bellman (HJB) equation which describes the optimal control problem for the evolution of the distribution of agents' states over time. In our work, we introduce mean-field game and mean-field control (MFC) into GNN, as described below.

A MFC framework for designing a novel GNN model. First, we extend the heat equation to a graph-based reaction-diffusion model, where the system behavior is determined by a mobility functional ($\Theta_1$) and reaction functional ($\Theta_2$). Second, we follow the recent work of MFC (Li et al., 2022) to define the main variational problem, consisting of a metric space (Eq. 6) and gradient flow (Eq. 7). After that, we study the critical point of the variational problem, yielding a Hamiltonian flow in MFC problem (Proposition 2).

Takeaway. We formulate the dynamic process of graph feature representation as a mean-field game where the game policy is defined in a mean-field control perspective. In the real application, our model simultaneously (1) crafts GNN model instance by identifying the most appropriate game policy (i.e., derive the instance of mobility functional $\Theta_1$ and reaction functional $\Theta_2$), and (2) optimizes GNN instance using Hamiltonian flow.

### A.6 ALGORITHM IMPLEMENTATION

---

**Algorithm 1:** *Nash-GNN* algorithm

---

**Input:** Graph $\mathcal{G} = (\mathcal{V}, \mathcal{P})$, node features $x(t)$, adjacency matrix $A$
**Output:** The mobilities $\Theta_1, \Theta_2$, the evolved node feature representation $x(T)$
**for** $i = 1 \ldots |\mathcal{V}|$ **do**
    Construct phase space by $(\Psi_i, \Phi_i) \leftarrow \mathcal{F}(x_i(t))$;
    **for** $t = 1 \ldots T$ **do**
        Learn mobilities $\Theta_1, \Theta_2$ by $\Theta_1, \Theta_2 \leftarrow \mathcal{M}((\Psi, \Phi))$;
        Construct Hamiltonian function $\mathcal{H}(\Psi_i(t), \Phi_i(t))$ by Eq. 10;
        Build PDE of the evolution of system state on graph by Eq. 12;
        Derive the trajectory $(\Psi_i(t), \Phi_i(t))$ by PDE solver;
    **end**
    Yield the evolved node feature representation $x(T)$ by $x(T) \leftarrow \Pi(\Psi(T), \Phi(T))$;
**end**

---

## B EXPERIMENTAL DETAILS

### B.1 DATASETS AND HYPERPARAMETERS

*Classic graph data for node classification.* We summarize the data information in the following Table 5.

*Classic graph data for graph classification.* We summarize the involved TUDdataset in the following Table 6.

Table 5: Data description for node classification.

| | Texas | Wisconsin | Actor | Squirrel | Chameleon | Cornell | Citeseer | Pubmed | Cora | ogbn-arxiv | ogbn-products |
|---|---|---|---|---|---|---|---|---|---|---|---|
| Hom. ratio $h$ | 0.11 | 0.21 | 0.22 | 0.22 | 0.23 | 0.3 | 0.57 | 0.74 | 0.81 | 0.23 | 0.41 |
| #Nodes $|\mathcal{V}|$ | 183 | 251 | 7,600 | 5,201 | 2,277 | 183 | 3,327 | 19,717 | 2,708 | 169,343 | 2,449,029 |
| #Edges $|\mathcal{P}|$ | 295 | 466 | 26,752 | 198,493 | 31,421 | 280 | 4,676 | 44,327 | 5,278 | 1,116,243 | 61,859,140 |
| #Classes $|\mathcal{Y}|$ | 5 | 5 | 5 | 5 | 5 | 5 | 7 | 3 | 6 | 23 | 27 |

Table 6: TUDataset description.

| | MUTAGE | NCI1 | ENZYMES | D&D | PTC_FM | IMDB | PROTEINS |
|---|---|---|---|---|---|---|---|
| #Graphs $|\mathcal{V}|$ | 188 | 4,110 | 600 | 1,178 | 349 | 1,000 | 1,113 |
| #Classes $|\mathcal{P}|$ | 2 | 2 | 6 | 2 | 2 | 2 | 2 |
| Avg#Nodes $|\mathcal{Y}|$ | 17.93 | 29.87 | 32.63 | 284.32 | 14.11 | 19.77 | 39.06 |
| Avg#Edges $|\mathcal{Y}|$ | 19.79 | 32.30 | 62.14 | 715.66 | 14.48 | 96.53 | 72.82 |

*Disease-based human connectome data*. We summarize the data information in the following Table 7. Note, Destrieux atlas (Destrieux et al., 2010) (160 brain regions) are used in OASIS to verify the scalability of the models.

Table 7: Disease-based human connectome data statistics.

| Dataset | Condition | # of Subjects | # of Classes | # of Regions/Nodes | Avg # of Node Features |
|---|---|---|---|---|---|
| **ABIDE** | Autism | 1025 | 2 | 116 | 201 |
| **ADNI** | Alzheimer | 250 | 5 | 116 | 177 |
| **OASIS** | Alzheimer | 1475 | 2 | 160 | 330 |
| **PPMI** | Parkinson | 209 | 4 | 116 | 198 |

For a binary dataset consisting of two classes, one representing a disease group and the other a normal control group. For ADNI dataset, following the clinical outcomes, we categorized subjects into distinct groups representing different cognitive statuses. These groups include: cognitive normal (CN), Subjective memory concern (SMC), early-stage mild cognitive impairment (EMCI), late-stage mild cognitive impairment (LMCI), and Alzheimer's Disease (AD) groups. To facilitate population counts, we regard CN, SMC and EMCI as "CN-like" group, while LMCI and AD as "AD-like" groups. This partitioning allows for the analysis and comparison of individuals across varying levels of cognitive function, providing valuable insights into disease progression and cognitive decline within the study population. For the PPMI dataset, which encompasses four distinct classes, including normal control, scans without evidence of dopaminergic deficit (SWEDD), prodromal, and Parkinson's disease (PD).

*Hyperparameters.* We use the Adam optimizer with a learning rate of 0.01, and the epoch is set as 250. Most hidden dimensions are set to 128 (Squirrel and Chameleon are set to 64, Cora and ABIDE are set to 32). All the experiments are conducted on four *NVIDIA RTX 6000* Ada GPUs. The code is released at Anonymous GitHub: `https://anonymous.4open.science/r/Nash-GNN-4570/`.

## B.2 COMPARISON METHODS

Graph neural networks (GNNs) have emerged as powerful tools for learning from graph-structured data, achieving state-of-the-art performance in various domains such as social networks, biological networks, and recommendation systems. In this work, we compare our method against a diverse set of benchmark GNN models that represent key advancements in the field:

Vanilla GCN (Kipf & Welling, 2017): The Graph Convolutional Network (GCN) introduced the foundational concept of convolutional operations on graph-structured data, leveraging spectral graph theory to propagate node features across the graph. Despite its simplicity, GCN remains a widely used baseline in GNN research.

GAT (Veličković et al., 2018): The Graph Attention Network (GAT) improved upon GCN by incorporating an attention mechanism to adaptively weigh neighbor contributions, enabling the model to capture more nuanced patterns in heterogeneous and large-scale graphs.

GraphSAGE (Hamilton et al., 2017b): This inductive framework generates embeddings by sampling and aggregating features from node neighborhoods, making it particularly effective for large and dynamic graphs where new nodes can be introduced.

GraphCON (Rusch et al., 2022): GraphCON leverages neural ODEs and skip connections to improve gradient flow during training, enabling it to address the oversmoothing problem in deep GNN architectures.

GraphBel (Song et al., 2022): This model focuses on enhancing robustness against adversarial attacks by learning more resilient graph representations through belief propagation mechanisms.

GRAND (Chamberlain et al., 2021): GRAND introduces random diffusion processes to improve message passing, focusing on long-range dependencies and reducing the oversquashing issue commonly seen in deep GNNs.

ACMP (Wang et al., 2022): The Adversarial Contrastive Message Passing (ACMP) framework utilizes contrastive learning to enhance node representations, particularly in the presence of noisy or incomplete graphs.

HANG (Zhao et al., 2024): HANG employs adversarial training to learn robust graph embeddings, effectively tackling challenges posed by graph perturbations and adversarial noise.

GIN (Xu et al., 2018): The Graph Isomorphism Network (GIN) is designed to be as powerful as the Weisfeiler-Lehman graph isomorphism test, achieving high expressiveness by using a learnable aggregation function.

AM-GCN (Wang et al., 2020): This model integrates both node features and graph topology in a balanced way, enhancing its ability to learn from graphs with highly diverse connectivity patterns.

These models collectively capture a wide range of design principles, from improved aggregation mechanisms (e.g., GAT, GIN) and inductive capabilities (e.g., GraphSAGE) to adversarial robustness (e.g., HANG, GraphBel) and advanced training techniques (e.g., GraphCON, GRAND). By benchmarking against these state-of-the-art GNNs, we provide a comprehensive evaluation of our method's performance, highlighting its strengths and contributions to the field.

Table 8: The performance on OGB dataset.

|  | GCN | GAT | GraphSAGE | GraphBel | ACMP | HANG | GCNII | AM-GCN | *Nash-GNN* |
|---|---|---|---|---|---|---|---|---|---|
| **ogbn-arxiv** | 0.7174 | 0.7365 | 0.7149 | 0.7256 | 0.7543 | 0.7484 | 0.7274 | 0.7239 | **0.7697** |
| **ogbn-products** | 0.7564 | 0.7904 | 0.7870 | 0.8049 | 0.8295 | 0.8468 | 0.7824 | 0.8013 | **0.8740** |

## B.3 ABLATION STUDY

Table 9: Ablation studies on Hamiltonian energy function $\mathcal{H}$ and PDE solver.

|  | **Texas** | **Wisconsin** | **Actor** | **Squirrel** | **Chameleon** | **Cornell** | **Citeseer** | **Pubmed** | **Cora** |
|---|---|---|---|---|---|---|---|---|---|
| **Hamiltonian energy function $\mathcal{H}$** | | | | | | | | | |
| GCN | 0.8715 | 0.8200 | 0.3403 | 0.4019 | 0.5522 | 0.7495 | 0.7075 | 0.8880 | 0.8739 |
| GAT | 0.9083 | 0.8466 | 0.3544 | 0.3506 | 0.5913 | 0.7266 | 0.7313 | 0.8895 | 0.8703 |
| **PDE Solver** | | | | | | | | | |
| Euler | 0.9266 | 0.8733 | 0.3529 | 0.4013 | 0.6043 | 0.8716 | 0.7663 | 0.9037 | 0.8832 |
| RK4 | 0.8073 | 0.8700 | 0.3912 | 0.4415 | 0.6565 | 0.8807 | 0.76627 | 0.8896 | 0.8867 |
| Symplectic-Euler | 0.8716 | 0.8600 | 0.3586 | 0.4013 | 0.6130 | 0.8165 | 0.7155 | 0.8769 | 0.8686 |
| dopri5 | 0.9541 | 0.8767 | 0.3917 | 0.3795 | 0.6522 | 0.8624 | 0.7394 | 0.8911 | 0.8467 |

*Energy function:* In physical systems, the system is often depicted as a graph where two neighboring vertices with mass are connected by a spring of given stiffness and length (Curtin & Scher, 1990). In this context, the system's energy is thus related to the graph's topology thus we design two examples of energy function $\mathcal{H}$ that involve interactions between neighboring nodes, one is vanilla GCN, and the other one is GAT.

We can observe that such simplification will degrade the performance compared with our *Nash-GNN*, whereas outperforms other competing methods in most scenarios (as shown in Tables 2, 8, 3, 4). In contrast, our *Nash-GNN* is not to directly learn the energy function, but to indirectly learn the Hamilton function by fitting the potential function (mobility) $\Theta$ through Eq. 10. By doing

so, the defined $\Theta$ has strict mathematical derivation. We use an Input Convex Neural Network (ICNN) to learn $\Theta$ because it guarantees that the objective function is convex, ensuring robustness and mathematical rigor in the model.

*PDE solver:* We list the performance of various PDE solvers in Table 9. The solvers considered include the fixed-step Euler and RK4 methods, the adaptive-step Dopri 5 method from (Chen et al., 2018), and the Symplectic-Euler method from (Rusch et al., 2022). The Symplectic-Euler method, being inherently energy-conserving, is particularly well-suited for preserving the dynamic properties of Hamiltonian systems over long periods. Our observations suggest that the choice of solver influences the performance of models. However, optimizing solver selection was not a major focus of our work, and no specific optimizations were performed during our experiments. For computational efficiency, we opted for the Euler PDE solver in all experiments presented in the main paper.

## C   DISCUSSION, LIMITATIONS AND SOCIAL IMPACT

*Discussion.*   In our experiments, we observed that the number of MLP layers in the ICNN module will impact performance. Specifically, smaller values of $h$ may require more MLP layers to adequately capture and model the underlying data. One plausible explanation for this phenomenon is that heterophilic graph data exhibits a more complex relationship between edges compared to homophilic graph data. This complexity necessitates deeper neural network architectures to effectively learn and represent the nuanced relationships present in the data.

*Limitations.* Our model has a relatively high computational cost. We list the average inference time for different models used in our study in Table 10. This analysis is performed using the Cora dataset (2708 nodes), with all graph PDE models employing the Euler Solver, an integration time of 3, and a step size of 1. All the experiments are conducted on NVIDIA RTX 6000Ada GPUs. Upon examination, it is observed that our *Nash-GNN* necessitates more inference time compared to other baseline models. But it has the same inference time as methods of the same type, such as GraphBel, ACMP and HANG.

| Model | GCN | GAT | GraphSAGE | GraphCON | GraphBel | GRAND | ACMP | HANG | GCNII | *Nash-GNN* |
|---|---|---|---|---|---|---|---|---|---|---|
| Time(ms) | 2.5570 | 4.5409 | 1.0505 | 2.2354 | 24.5252 | 10.228 | 27.1618 | 33.4651 | 3.0198 | 33.2494 |

Table 10: Model inference time (ms) comparison across various architectures.

*Societal impact.*   Our major contribution to the machine learning field is we introduce a principled approach to optimize GNNs for suiting diverse graph datasets. Through the integration of mean-field control theory and Hamiltonian flows into GNN abstract learning, we developed a novel methodology that enhances our understanding of deep learning models applied to graph datasets. From the application perspective, our deep model represents a promising approach to bridge the gap between graph-based machine learning and neuroscience research, offering new avenues for studying disease processes.

