# OpenReview forum: "Graph Neural Network Is A Mean Field Game"
_ICLR.cc/2025/Conference — ICLR 2025 Conference Withdrawn Submission_

### Official Review · Reviewer_ihdE · 2024-10-28

**Soundness:** 3
**Presentation:** 2
**Contribution:** 4
**Rating:** 6
**Confidence:** 2

**Summary:**

The authors propose a new framework, that investigates GNNs from the perspective of mean field theory. Within this framework, they propose a new model architecture called NashGNN, which is reported to achieve better than state of the art performance on node and graph classification on  several benchmark datasets.

**Strengths:**

1) The proposed algorithm surpasses the state of the art.

2) The framework is novel.

3) The authors made extensive experiments.

**Weaknesses:**

1) The paper appears not to be written for the ML/GNN community.

a) The introduction is rather long and covers GNN downsides, that are not mentioned in the rest of the paper.

b) I believe the motivation is that the proposed framework can offer a solution for all learning tasks an data sets, but this motivation (or any other) is not given precisely enough in the introduction.

c) The concept of mean field theory and why it can be used for GNNs is not well explained for reader with ML background.

d) Some important information can only be found in the appendix or footnotes: the pseudo code is in the appendix, the eq that tells me how the graph topology is used in the model is in footnote 2, the explanation for Def 1 is in the appendix, model limitations are briefly mentioned in the appendix

2) The authors claim in the intro, that state of the art performance is not well suited for the underlying graph problem (p2., line 89-97) but evaluate their algorithm comparing it to state of the art.

3) The authors claim to be the first to propose a graph meta-learning method (p. 7, line 330), but I found several other papers, for example: https://dl.acm.org/doi/abs/10.1145/3357384.3358106  and https://ieeexplore.ieee.org/abstract/document/9772740

4) The GIN model is missing in the benchmark algorithms.

5) The read thread can be improved.

6) There is no comprehensive state of the art section.

7) Figure 1 does not really help in understanding the method and figure 3 is confusing.

**Questions:**

a) Can you please explain, where exactly the graph structure is used in the model, is it only as specified in footnote 2?

b) I would suggest shortening the intro and making it more concise.  Specifically, you can shorten the part on GNN downsides but make the motivation more clear. Why do you regard the node feature vectors as potential energy?

c) I believe an introductory paragraph on mean field theory as well as, short explanatory sentences instead of remarks would improve the red thread.

d) Can you please clarify or elaborate on what you meant with the sentence I mentioned under weakness 2?

e) Can you please elaborate on your claim I mentioned in weakness 3? I believe for meta learning you should have experiments, where you actually learned on one task and can still predict in another.

f) Can you please compare your model also to the GIN model, since it is maximally expressive.

g) The discussion on page 10 seams to draw very strong conclusions regarding neuro-degenerative diseases. Please either provide more supporting investigations or formulate less strongly.

---

> ### Author Response · Authors · 2024-11-23
> **Response to Reviewer ihdE**
>
> ### Thank you for acknowledging the contributions of our work. We are thrilled and grateful for your insightful feedback, which has significantly contributed to enhancing the quality of our manuscript. In the following responses, **W** and **Q** represent Weaknesses and Questions, and **A** represents the corresponding answer. ###
>
> **W1a:** The paper appears not to be written for the ML/GNN community.
>
> **A1a:**: Thank you for your constructive comment. We have reorganized the introduction. Since our overarching goal is to unify the existing GNN models in the MFC framework, we intended to make the introduction section self-contained by covering the learning principle and current challenges. Due to the page limit, a large portion of the experiment results have been moved to the Appendix.
>
> Please refer to the revised manuscript.
>
> **W1b:** I believe the motivation is that the proposed framework can offer a solution for all learning tasks an data sets, but this motivation (or any other) is not given precisely enough in the introduction.
>
> **A1b:**: Thank you for your constructive comment. One of our major motivations is to overcome the bias of any assumption on the characteristics of graph topology. To achieve it, we formulate GNN as a mean-field game and introduce MFC theory to use the best game policy for the unseen graph data. The state-of-the-art performance on both homophilic and heterophilic data (shown in Table 2) underscores the advantage of our MFC framework for designing new GNN models. We have clarified the motivation of our work in the revised manuscript (please refer to Sec. 2.1).
>
> **W1c:**: The concept of mean field theory and why it can be used for GNNs is not well explained for reader with ML background.
>
> **A1c:**: Thank you for your constructive comment, due the limit page, we added a detailed explanation in the original manuscript for why GNN is a mean file game in Appendix A.5, and Figure 4 shows the evolution of our Nash-GNN from GNN. For clarity, we have added a clear explanation in the revised manuscript, please refer to Sec 2.4 (line 316 to line 328).
>
> **W1d:**: Some important information can only be found in the appendix or footnotes
>
> **A1d:**: Thank you for your constructive comment, we removed some important information into the main manuscript, please refer to the highlighted part in the updated manuscript.
>
> **W2 \& Q4:**  The authors claim in the intro, that state of the art performance is not well suited for the underlying graph problem (p2., line 89-97) but evaluate their algorithm comparing it to state of the art.
>
>   **A2:**: Thank you for your thoughtful comment. Our intention was to validate this claim empirically. Therefore, we conducted a comprehensiveevaluation of our algorithm by comparing it against the current state-of-the-art methods to highlight the differences and limitations in existing approaches.
>
> **W3 \& Q5:**   The authors claim to be the first to propose a graph meta-learning method (p. 7, line 330), but I found several other papers
>
>  **A3:**: We apologize for the confusion. Our major contribution is a novel MFC framework to design novel GNN models for unseen graph data. We are aware of existing work on meta-learning on graph data. Therefore, we are not going to claim that our method is the first effort to explore graph meta-learning. Instead, we simply highlight in the text that our work has a meta-learning flavor. We have carefully reviewed our current write-up to make sure there is no such indication related to the innovation of meta-learning.
>
> **W4 \& Q6:**   The GIN model is missing in the benchmark algorithms.
>
>  **A4:** Thank you for your comment, we have added GIN as the benchmark algorithm (please refer to Table 3 and Appendix B.2). The data splitting method follows this work https://arxiv.org/abs/1911.07979.
> | Dataset    | MUTAG       | NCI1        | PTC_FM      | IMDB-B      | PROTEINS    | D&D         | ENZYMES      |
> |------------|-------------|-------------|-------------|-------------|-------------|-------------|--------------|
> | GIN   | 81.39 ± 1.53 | 75.00 ± 1.40 | 59.00 ± 3.30 | 72.78 ± 0.86 | 71.46 ± 1.66 | 73.00 ± 3.30 | 49.60 ± 4.50 |
>
> **W5:** The read thread can be improved.
>
>  **A5:**: Thank you for your comment. We have rephrased the hard part according to all reviewers’ comments. Please refer to the marked parts in the revised manuscript.
>
> **W6:** There is no comprehensive state-of-the-art section.
>
>  **A6:**: Thank you for your valuable comment. We have added a comprehensive state-of-the-art section, please refer to Appendix B.2.
>
> **W6:** Figure 1 does not really help in understanding the method and figure 3 is confusing.
>
>  **A7**: Thank you for this comment. We have improved Fig. 1 and Fig. 3, and we have added more explanation for Fig. 3, please refer to Fig.1 and Fig. 3 in the updated manuscript.

---

> > ### Author Response · Authors · 2024-11-23
> > **Response to Reviewer ihdE**
> >
> > **Q1:** Can you please explain, where exactly the graph structure is used in the model, is it only as specified in footnote 2?
> >
> > **A8:** Thank you for your comment. Like other GNN models, our model uses graph convolutions to propagate information across the nodes and edges of the graph.
> >
> >  **Q2:** I would suggest shortening the intro and making it more concise. Specifically, you can shorten the part on GNN downsides but make the motivation more clear. Why do you regard the node feature vectors as potential energy?
> >
> > **A9:** Thank you for your valuable comment. We have shortened the introduction part and the motivation explanation in Sec. 2.1 (please refer to line 152 to line 160).
> >
> >  **Q3:**  I believe an introductory paragraph on mean field theory as well as, short explanatory sentences instead of remarks would improve the red thread.
> >
> > **A10:** We appreciate this constructive comment. We have taken this into the account in the rewritten part.
> >
> >  **Q7:** The discussion on page 10 seams to draw very strong conclusions regarding neuro-degenerative diseases. Please either provide more supporting investigations or formulate less strongly.
> >
> >  **A11:** Thank you for your comment. We have rephrased it. Please refer to line 515 to line 528.

---

> ### Author Response · Authors · 2024-12-01
> **Looking forward to further engaging in the discussion!**
>
> Dear, Reviewer ihdE,
>
> We sincerely appreciate your time and effort in reviewing our manuscript and providing such constructive feedback. As the author-reviewer discussion phase is nearing its end, we wanted to check if you have any further comments or questions regarding our responses. We would be more than happy to continue the conversation if needed. If we have addressed your concerns, would you like to update your rating?
>
> Thank you so much.
>
> Best,
>
> Authors

---

> > ### Comment · Reviewer_ihdE · 2024-12-02
> > **Reply to Revision**
> >
> > Dear authors,
> > I very much appreciate, you adding GIN to your baseline. However I still have concerns regarding the figures, which is why I prefer to keep my original rating.

---

> ### Author Response · Authors · 2024-12-02
> **We really appreciate your constructive comment.**
>
> Dear, Reviewer ihdE,
>
> We really appreciate your constructive comment. We are trying our best to address your concerns promptly.
>
> - For Fig.1, we have redrawn it and uploaded it at https://anonymous.4open.science/r/Nash-GNN-4570/Fig1_revised.png.
>
> - For Fig. 3, we have redrawn it and uploaded it at https://anonymous.4open.science/r/Nash-GNN-4570/Fig3_revised.png. Also, we can replace it by using the pseudo-code (please refer to Appendix A.6).
>
> We greatly appreciate your suggestion and assure you that it will be incorporated into the final version.
>
> Thank you very much for your valuable contribution!
>
> We will make every effort to thoroughly address all of your concerns. We sincerely hope you can reconsider your rating.
>
> Best,
>
> Authors

---

### Official Review · Reviewer_EZdm · 2024-11-02

**Soundness:** 2
**Presentation:** 1
**Contribution:** 2
**Rating:** 5
**Confidence:** 2

**Summary:**

The work formulates GNNs as a mean field game. My understanding is that the goal is to find a good trade off between information exchange (what the authors call transportation mobility) across the graph and possibly the sharpness of the features (what the authors call reaction mobility). The authors propose Nash-GNN to achieve this.

**Strengths:**

The authors present an interesting perspective on learning on graphs. I believe that there seem to be some fundamentally novel and interesting ideas within the work.

I liked Table 1 that acts as a unification between different existing variations types of methods that exist in the GNN literature. Overall I think that sections 2.1, 2.2, and 2.3 have a lot of potential, but they need to be better presented.

**Weaknesses:**

I found the paper to be very hard to understand, even though I’m familiar with many of the related works. I think the language in many cases is quite confusing and overall the paper lacks clarity.

As an example in the abstract the authors write “to jointly carve the nature of the inductive bias and fine-tune the GNN hyper-parameters on top of the elucidated learning mechanisms”. Phrases like this seem scattered throughout the work, which in my opinion are very hard to understand and make the paper challenging to read. There seems to be an overall distinct lack of clarity in the work.

Figure 1 is quite non-standard presenting a rather imprecise analogy.

Figure 2 is quite confusing and dense, it might be worth swapping Figure 2 with algorithm 1 in the appendix as that to me is much more clear.

Figure 3 is quite confusing, the caption needs more information.

I found Table 5 to be misleading as Nash-GNN has results in bold for 100% for the TaoWu, PPMI, and Neurocon datasets when many techniques seem to achieve 100% (so at the very least they should be coloured in bold as well).

Many terms are poorly defined such as “transportation mobility” and “reaction mobility”, It would be quite important in my opinion to change these terms to make them more digestible for someone working in the field of GNNs.

**Questions:**

Please see weaknesses.

Overall, I believe that the ideas contained in the paper are interesting and worthy of a paper, but the presentation unfortunately detracts significantly from the readability of the work. My recommendation is that the authors take time to re-write the paper in order to make the main message much more clear. I am however keen to read the opinions of the other reviewers to see if I have missed something.

To me it was also unclear exactly how Nash-GNN differs from for example GRAND. It would be useful to compare the models more concretely, I.e. I think that Nash-GNN should appear in Table 1 to make it more clear what the difference is.

---

> ### Author Response · Authors · 2024-11-22
> **Response to Reviewer EZdm**
>
> ### Thank you for acknowledging the contributions of our work. We are thrilled and grateful for your insightful feedback, which has significantly contributed to enhancing the quality of our manuscript. In the following responses, **W** and **Q** represent Weaknesses and Questions, and **A** represents the corresponding answer. ###
>
>  **W1:** ...the paper to be very hard to understand...
>
> **A1:** We apologize for the confusion. We reviewed and restructured the storyline of our work, Specifically, we have added a “Background and Motivation” section (please refer to line 151 to line 176 in the updated manuscript) to highlight the significance of the variational framework of mean-field control as a novel approach for developing innovative GNN models.
> The structure of the "2 Methods" section has been reorganized, as illustrated in the right panel of the updated Fig. 1. First, we frame the feature representation learning in GNN as a dynamical system of minimizing an energy function via a pre-defined gradient flow. Second, we extend the conventional graph diffusion process to a reaction-diffusion model, yielding a general mean-field control framework that unities most of the current GNN works. Furthermore, we reformulated the functional $\Theta_1$ and $\Theta_2$ as the control pattern in the mean field, which allows us to reshape the message-passing mechanism in GNNs. By learning the control patterns from the underlying graph data, we are able to jointly identify the most suitable learning mechanism and fine-tune the GNN hyper-parameters in a “meta-learning” paradigm.
>
> **W2:** ...There seems to be an overall distinct lack of clarity in the work...
>
> **A2:**  We have rephrased and rewritten almost 30% of the write-up. We appreciate the valuable feedback that allows us to significantly improve the clarity of our work. Please refer to the marked BOLD line parts in the updated manuscript.
>
> **W3:** Figure 1 is quite non-standard presenting a rather imprecise analogy.
>
> **A3:** We partially agree with this reviewer that the cartoon in Fig. 1 might not be conventional in a technical conference. However, we respectfully argue that the analogy exactly highlights the limitations of current GNN models and clearly conveys our proposed solution to address this challenge. The page limit is partly responsible for this confusion, so allow us to provide a more detailed explanation in this rebuttal. If the reviewer finds it helpful, we will incorporate it into the final version.
>
> The example of gearbox is used to illustrate the issue that the current GNN model applies the fixed learning mechanism (same gear) to all graph data. While current GNN models demonstrate promising learning performance, it is highly likely that their effectiveness stems more from data-fitting capabilities than from a good understanding of the graph learning mechanism, making them prone to overfitting. Take the gearbox in the manual transmission car as an example. Although it is mechanically possible to drive on the highway using low gears, it is uncommon because, over time, it can cause damage to the engine. In this context, the ability to shift gears according to different traffic conditions is highly desirable. Similarly, we propose to incorporate an (outer-loop) learning component, on top of the (inner-loop) GNN optimization, to identify the most suitable learning mechanism by shaping the control patterns $\Theta_1$ and $\Theta_2$.
>
> **W4:** Figure 2 is quite confusing and dense, it might be worth swapping Figure 2 with algorithm 1 in the appendix as that to me is much more clear.
>
> **A4:** We appreciate this constructive comment. We will swap Fig. 2 and the table of Algorithm 1 in the final version.
>
> **W5:** Figure 3 is quite confusing, the caption needs more information.
>
> **A5:** We apologize for the confusion. We have updated the plotting of Fig. 3 as well as the caption. Please check the highlighted part in the new version. Thanks.

---

> > ### Author Response · Authors · 2024-11-22
> > **Response to Reviewer EZdm**
> >
> > **W6:** I found Table 5 to be misleading as Nash-GNN has results in bold for 100% for the TaoWu, PPMI, and Neurocon datasets when many techniques seem to achieve 100% (so at the very least they should be coloured in bold as well).
> >
> > **A6:** Thank you for your comment. After reviewing the data and model, we decided to take TaoWu (for AD) and Neurocon (for PD) datasets out of the experiment, as we are concerned that the small sample size (40 subjects) could introduce bias into the classification results. Meanwhile, we replace the binary classification (CN vs PD) with a 4-category classification (CN, SWEDD, prodrome PD, PD) in PPMI dataset, since the research focus of computer-assisted diagnosis is to shift the diagnosis window to the early stage (with subtle brain alterations). Please refer to Table 5 in the updated revised manuscript.
> >
> > *Note: CN refers to cognitive normal, SWEDD refers to Scans Without Evidence of Dopaminergic Deficit, and prodrome PD refers to the early, often subtle symptoms and signs that appear before the classic motor symptoms of Parkinson's disease (PD) become evident.*
> >
> > **W7:** Many terms are poorly defined such as “transportation mobility” and “reaction mobility”, It would be quite important in my opinion to change these terms to make them more digestible for someone working in the field of GNNs.
> >
> > **A7:** We appreciate this feedback. In line with the mean-field control framework, we define $\Theta_1$ and $\Theta_2$ as the control patterns corresponding to the diffusion and reaction components in the RDM. Additionally, we have restructured the "Methods" section to enhance clarity in the presentation.
> >
> > **Q1:** Overall, I believe that the ideas contained in the paper are interesting and worthy of a paper, but the presentation unfortunately detracts significantly from the readability of the work. My recommendation is that the authors take time to re-write the paper in order to make the main message much more clear. I am however keen to read the opinions of the other reviewers to see if I have missed something.
> >
> > **A8:**  We sincerely apologize for the issue with the clarity of the presentation. We have rewritten approximately 30% of the text, focusing on restructuring the storyline without introducing new content. Please refer to the highlighted sections in the revised version.
> >
> > **Q2:** To me it was also unclear exactly how Nash-GNN differs from for example GRAND. It would be useful to compare the models more concretely, I.e. I think that Nash-GNN should appear in Table 1 to make it more clear what the difference is.
> >
> > **A8:** We thank for these valuable comments. Below are our point-to-point responses.
> >
> > $\underline{\text{Regarding how Nash-GNN differs from GRAND, we briefly list the following major differences.}}$ (1)   GRAND (and other models) use the fixed diffusion function $F$, reaction function $R$ and the control patterns. However, our Nash-GNN learns the most suitable functions, on the fly, for each graph data. This explains why we can not write down the specific functions in Table 1 for our method. (2) Most of the current GNN models only focus on the diffusion process. In contrast, the gradient flow in our Nash-GNN is characterized by a more general reaction-diffusion model. (3) Our method integrates an outer-loop learning component (to identify the control pattern from the graph data) on the GNN model instance, resulting in a "meta-learning" approach. (4) Our Nash-GNN provides a new window to understand the learning mechanism on each graph node, as the promising results are shown in Fig. 3, which complement the widely used graph attention approach.

---

> ### Comment · Reviewer_EZdm · 2024-11-23
> **Thanks!**
>
> I would like to thank the authors for their large effort in this rebuttal. Reading the reviews from ihDe, I tend to agree with their point that the paper does not seem to be written for a “GNN audience”. I would also like to acknowledge the review from aaZh, where they mention that the “paper was clear and easy to follow” — something which I personally disagree with. I believe that the new revision does increase its readability, but the amount of changes make me wonder if another review cycle would be needed. Perhaps the AC can comment on this.
>
> I am not opposed to the acceptance of this work, but I do not believe that I have understood it well enough to provide a definite accept score. For this reason I would like to increase my score, but keep my confidence at a 2. This is not a comment on the technique but rather mostly on the presentation. I do believe sections such as 2.3.1 are in the right direction, but in my opinion still more work is needed to increase the target audience of this work.

---

> > ### Author Response · Authors · 2024-11-23
> > **We appreciate your valuable feedback!**
> >
> > Dear Reviewer EZdm,
> >
> > Thank you for your timely feedback and for considering our rebuttal. We appreciate your acknowledgment of the improvements in readability. Regarding the rephrasing and revisions, we want to clarify that most of the changes involved reorganization and the addition of explanations to enhance clarity and accessibility for the target audience. Importantly, the core contributions and innovations of the paper remain unchanged. We also value your perspective on increasing the reach of this work to a broader GNN audience and will continue to refine sections like 2.3.1 to address this. Your feedback is invaluable in guiding these efforts.
> >
> > Have a great weekend.
> >
> > Best,
> >
> > Authors

---

### Official Review · Reviewer_aaZh · 2024-11-05

**Soundness:** 2
**Presentation:** 2
**Contribution:** 2
**Rating:** 5
**Confidence:** 3

**Summary:**

This paper proposes a variational framework for abstract learning in Graph Neural Networks (GNNs) from a mean-field control perspective. The authors demonstrate that several existing GNN architectures can be viewed as specific instances within this framework, differing in their choice of diffusion and reaction functions. Building upon this framework, they design an end-to-end deep learning architecture that characterizes information propagation in GNNs. Experimental results on benchmark graph datasets show that the proposed framework significantly outperforms GNN baselines in both node and graph classification tasks.

**Strengths:**

1. The paper is clear and easy to follow.

2. The proposed framework is interesting, and several existing graph neural diffusion architectures fit into this framework, even though viewing GNNs as dynamic systems has been studied in the literature (e.g., [1]).

3. The experimental results on OGBN and TUDatasets demonstrate that the proposed model significantly outperforms several GNN baselines.

**Weaknesses:**

1. While the proposed framework is intriguing, it does not address several important challenges such as over-smoothing, over-squashing, and the ability to capture long-range dependencies in graphs. As a result, it is unclear whether the framework can mitigate these problems.

2. Some of the datasets used in the experiments are either relatively small (e.g., Texas, Wisconsin, and Cornell, which have only several hundred nodes) or problematic. For instance, as highlighted in Section 3.1 of [2], the Chameleon and Squirrel datasets reportedly contain repeated nodes, casting doubt on their reliability for fair and robust model testing.

3. It would be good to clarify the conditions required for the optimization problem in Eq. (6) to be well-posed, i.e., to have a unique solution. Moreover, when Eq. (6) is non-convex, how does converging to a local minimum influence the model's predictive performance?


**References:**

[1] Di Giovanni, F., Rowbottom, J., Chamberlain, B. P., Markovich, T., & Bronstein, M. M. (2023). Understanding Convolution on Graphs via Energies. TMLR.

[2] Platonov, O., Kuznedelev, D., Diskin, M., Babenko, A., & Prokhorenkova, L. (2023). A Critical Look at the Evaluation of GNNs under Heterophily: Are We Really Making Progress?. ICLR.

**Questions:**

1. Under what scenarios would non-linear diffusion and reaction functions be superior to linear functions, and vice versa? Could you provide some insights on this?

2. Could you conduct evaluations on the filtered versions of the Chameleon and Squirrel datasets, as well as on new heterophilic datasets provided by Platonov et al. [2]?

---

> ### Author Response · Authors · 2024-11-23
> **Response to Reviewer aaZh**
>
> ### Thank you for acknowledging the contributions of our work. We are thrilled and grateful for your insightful feedback, which has significantly contributed to enhancing the quality of our manuscript. In the following responses, **W** and **Q** represent Weaknesses and Questions, and **A** represents the corresponding answer. ###
>
> **W1**: While the proposed framework is intriguing, it does not address several important challenges such as over-smoothing, over-squashing, and the ability to capture long-range dependencies in graphs. As a result, it is unclear whether the framework can mitigate these problems.
>
> **A1**: Thank you for your constructive comments from an application perspective. In a high-level pitch, we conjecture that one possible solution to address the common issues in GNN, such as over-smoothing and over-squashing, is to use the appropriate learning mechanism in information exchange between connected graph nodes. For instance, in homophilic graph data, neighboring nodes are encouraged to have similar feature representations. Otherwise, in heterophilic graphs, information exchange should be discouraged, even if the connectivity is strong. In this regard, our major contribution is to present a novel learning paradigm by conceptualizing the feature representation learning in GNN as a mean field game. By capitalizing on the variational framework and physical principles, we proposed to learn the optimal control pattern that allows us to achieve state-of-the-art performance by using the most effective learning mechanism for the unseen graph data.
>
> Although we have not specifically evaluated the effectiveness of addressing the over-smoothing issue in our original submission, we demonstrate the learned control patterns in Fig. 3 left, where we found the learned control pattern is highly correlated with the graph homophily ratio. In addition, we have conducted an extra experiment to demonstrate that our method enables the use of a much deeper network architecture by mitigating the over-smoothing issue. The results are shown as follows. We will include this result in the final version if this reviewer thinks it is necessary.
>
> |   Number of layers        | 1       | 2       | 8       | 16      | 32      | 64       | 128     |
> |-----------|---------|---------|---------|---------|---------|----------|---------|
> | Texas     | 0.92591 | 0.90037 | 0.89037 | 0.9185  | 0.88333 | 0.90740  | 0.89185 |
> | Wisconsin | 0.84000 | 0.85333 | 0.88667 | 0.88461 | 0.87333 |  0.88266 | 0.87615 |
> | Citeseer  | 0.76188 | 0.77313 | 0.74627 | 0.78507 | 0.75702 | 0.77611  | 0.78209 |
>
> In addition, unlike traditional GNNs, directly solving PDEs in our method enables fine-grained control over the speed and range of information propagation, allowing for more selective and localized information flow. This control helps prevent excessive information aggregation, ensuring that node representations do not neutralize too quickly, thus mitigating the over-smoothing problem.
>
> **W2**: Some of the datasets used in the experiments are either relatively small
>
>   **A2:** Thank you for this valuable comment. We have added two node property predictions (single-label classification) in OGB dataset (i.e., ogbn-arxiv and ogbn-products). We still keep the results of Texas, Wisconsin, and Cornell since they have been reported in many other GNN works.
>
> |               | GCN    | GAT    | GraphSAGE | GraphBel | ACMP   | HANG   | GCNII  | AM-GCN | Nash-GNN |
> | ------------- | ------ | ------ | --------- | -------- | ------ | ------ | ------ | ------ | -------- |
> | **ogbn-arxiv**    | 0.7174 | 0.7365 | 0.7149    | 0.7256   | 0.7543 | 0.7484 | 0.7274 | 0.7239 | 0.7697   |
> | **ogbn-products** | 0.7564 | 0.7904 | 0.7870    | 0.8049   | 0.8295 | 0.8468 | 0.7824 | 0.8013 | 0.8740   |
>
> We can observe that our *Nash-GNN* can achieve considerable results, we have added these results in the revised manuscript (please refer to Table 8).
>
> **W3**: It would be good to clarify the conditions required for the optimization problem in Eq. (6) to be well-posed, i.e., to have a unique solution. Moreover, when Eq. (6) is non-convex, how does converging to a local minimum influence the model's predictive performance?
>
>  **A3**: Thank you for your constructive comment. In our formulation, ensuring that Eq. 6 is a convex function is a necessary condition for identifying the critical point. This is why we utilize an Input Convex Neural Network (ICNN) to model the function.

---

> > ### Author Response · Authors · 2024-11-23
> > **Response to Reviewer aaZh**
> >
> > **Q1**: Under what scenarios would non-linear diffusion and reaction functions be superior to linear functions, and vice versa? Could you provide some insights on this?
> >
> >  **A4**: This is a very insightful question. We’d like to provide some of our understanding as follows.
> > In general, a graph learning task is essential to find a mapping function from the graph data tooutcomes/lables. Due to the complexity of real-world data, such mapping functions are often non-linear, as evidenced by the critical role of nonlinear operations like ReLU in neural networks. Since we formulate the feature representation in GNN as the dynamic system, using a non-linear model such as RDM is expected to be more powerful in capturing the complex relationship between and labels.
> > In addition, mounting evidence shows biological systems, such as human brain, are highly distributed and exhibit remarkable self-organized patterns. In the application of biomedical data, non-linear RDM is more suitable than simple linear models to characterize the growth trajectory on each node and node-to-node interactions.
> >
> > **Q2**: Could you conduct evaluations on the filtered versions of the Chameleon and Squirrel datasets, as well as on new heterophilic datasets provided by Platonov et al. [2]?
> >
> >  **A5**: Absolutely. We have performed the experiments on the filtered versions of the Chameleon and Squirrel datasets and new heterophilic datasets provided by Platonov et al. [2]. It is clear that our method consistently achieves promising results. If the reviewer thinks it necessary, we will incorporate it into the final version.
> >
> > |           | Chameleon      | Squirrel      | roman-empire | amazon-ratings | minesweeper   | tolokers     | questions    |
> > |-----------|----------------|---------------|--------------|----------------|---------------|--------------|--------------|
> > | GCN       | 40.89 ± 4.12   | 39.47 ± 1.47  | 73.69 ± 0.74 | 48.70 ± 0.63   |  89.75 ± 0.52 | 83.64 ± 0.67 | 76.09 ± 1.27 |
> > | GAT       | 39.21 ± 3.08   | 35.62 ± 2.06  | 80.87 ± 0.30 | 49.09 ± 0.63   | 92.01 ± 0.68  | 83.70 ± 0.47 | 77.43 ± 1.20 |
> > | GraphSAGE |   37.77 ± 4.14 | 36.09 ± 1.99  | 85.74 ± 0.67 | 53.63 ± 0.39   | 93.51 ± 0.57  | 82.43 ± 0.44 | 76.44 ± 0.62 |
> > | NashGNN   | 43.09 ± 0.69   | 39.71 ± 0.96  | 89.56 ± 0.35 | 55.94 ± 0.31   | 93.66 ± 0.43  | 85.09 ± 0.22 | 81.89 ± 0.13 |

---

> ### Author Response · Authors · 2024-11-30
> **Kindly Reminder**
>
> Dear, Reviewer aaZh,
>
> We sincerely appreciate your time and effort in reviewing our manuscript and providing such constructive feedback. As the author-reviewer discussion phase is nearing its end, we wanted to check if you have any further comments or questions regarding our responses. We would be more than happy to continue the conversation if needed. If we have addressed your concerns, would you like to update your rating?
>
> Thank you so much.
>
> Best,
>
> Authors

---

### Author Response · Authors · 2024-11-22
**General response to all reviewers**

We sincerely thank all reviewers for their thoughtful and constructive feedback. We have addressed all concerns raised by the reviewers (**marked by BOLD line in the updated manuscript**) and kindly request that the reviewers consider our rebuttals.

Also, we appreciate the recognition of our contributions and experimental rigor. Notably, all reviewers highlighted:

**(1) Technical Novelty**

- The proposed framework is **interesting**, and several existing graph neural diffusion architectures fit into this framework. (**Reviewer aaZh**)
- The proposed framework is **intriguing** (**Reviewer aaZh**)
- The authors present an **interesting perspective** on learning on graphs. I believe that there seem to be **some fundamentally novel and interesting ideas** within the work. (**Reviewer EZdm**)
- The framework is **novel**.  (**Reviewer ihdE**)

**(2) Comprehensive Experiments**

- The experimental results on OGBN and TUDatasets demonstrate that the proposed model **significantly outperforms** several GNN baselines. (**Reviewer aaZh**)
- I liked Table 1 that acts as a **unification** between different existing variations types of methods that exist in the GNN literature. Overall I think that sections 2.1, 2.2, and 2.3 have a lot of **potential**. (**Reviewer EZdm**)
- The proposed algorithm **surpasses** the state of the art. (**Reviewer ihdE**)
- The authors made **extensive** experiments. (**Reviewer ihdE**)

Reviewer aaZh noted, "The paper is **clear and easy to follow**."

In response, we have reorganized the paper further to enhance readability and ensure an even smoother understanding for our readers.

Thank you for your time and consideration.

Best,

Authors

---

> ### Author Response · Authors · 2024-11-27
>
> Dear Reviewers,
>
> We hope that our rebuttal has addressed your concerns and now we provide more clarity and understanding of our work in light of your valuable feedback.
>
> We sincerely appreciate your thoughtful comments and invite you to reach out with any additional questions or requests for clarification.
>
> We hope to contribute to the esteemed ICLR and the broader "learning on graphs and other geometries & topologies" community by having this paper published.
>
> Thank you for your time and consideration.
>
> Best,
>
> Authors

---

### Note · Authors · 2025-01-22

I have read and agree with the venue's withdrawal policy on behalf of myself and my co-authors.